# Quantifying cascading impacts through road network analysis in an insular volcanic setting: the 2021 Tajogaite eruption of La Palma Island (Spain)

Lucia Dominguez[1], Sébastien Biass[1], Corine Frischknecht[1], Alana Weir[1], Maria Paz Reyes-Hardy[1], Luigia Sara Di Maio[1], Nemesio Pérez[2,3], and Costanza Bonadonna[1]

[1]Department of Earth Sciences, University of Geneva, Geneva, 1205, Switzerland
[2]Instituto Volcanológico de Canarias (INVOLCAN), San Cristóbal de La Laguna, 38320, Spain
[3]Instituto Tecnológico y de Energías Renovables (ITER), Granadilla de Abona, 38600, Spain

*Correspondence to*: Lucia Dominguez (lucia.dominguez@unige.ch)

**Key words.** Post-event impact assessment, cascading impacts, network analysis, road connectivity, monogenetic volcanism

**List of abbreviations**

| | |
|---|---|
| Critical Infrastructure | CI |
| Damage and Disruption States | DDS |
| *Dirección General de Tráfico* (General Directorate of traffic) | DGT |
| Edge Betweenness Centrality | EBC |
| Edge Closeness Centrality | ECC |
| Hazard Intensity Metrics | HIM |
| Impact States | IS |
| Lower Unit of tephra deposit | LU |
| Mass Eruption Rate | MER |
| Medium Unit of tephra deposit | MU |
| OpenStreetMap | OSM |
| Origin-Destination routes | O-to-D |
| Post-Event Impact Assessment | PEIA |
| Upper Unit of tephra deposit | UU |

**Abstract**

Post-event impact assessments (PEIA) are essential to elucidate disasters' drivers and better anticipate future events. The 2021 Tajogaite eruption of Cumbre Vieja (La Palma, Spain) demonstrated the various orders of impact due to compound volcanic products (i.e., lava, tephra, gas) affecting a highly interconnected and low redundant infrastructure, typical of insular environments. Using a forensic approach, we discretise the causal order of cascading impacts, from physical damage (first order) to loss of functionality of the road network (second order) and subsequent systemic disruption of emergency management and socio-economic sectors (third order). Based on graph theory, we apply a comprehensive network analysis to quantify the loss of functionality and resulting effects, based on the spatiotemporal evolution of centrality indicators. The consequences on dependent systems are expressed in terms of increased driving time syn- and post-eruption between target locations for emergency (evacuation), public health (hospital), agriculture (crops-market), and education (schools). Graph indicators are objective measures of system performance during (disturbing/degraded states) and after the eruption (restorative state), when two new roads where rapidly built to reconnect the island. This study demonstrates how network analyses, informed by comprehensive PEIA, can accurately capture complex systemic disturbances, thus highlighting its potential for risk assessments.

## 1 Introduction

The functionality of modern society relies on an intricate configuration of socio-economic and political sectors that depend on the continuous operation of Critical Infrastructure (CI) systems (e.g., road, electricity, water networks). This delicate equilibrium becomes especially crucial when natural hazards disrupt decision-making, response and recovery capacity. Volcanic eruptions pose a unique challenge, as they can generate simultaneous phenomena and products (e,g, lava flows, pyroclastic flows, tephra fallout, gas) that provoke various orders of impact across diverse temporal and spatial scales. The simultaneous occurrence of hazards amplifying impacts is defined here as compound hazards (Cutter, 2018; Nemeth et al., 2024). However, volcanic impacts are amplified not only by the intensity and interaction of compound hazards, but also by their interaction with interconnected CI, producing nonlinear chains of cascading impacts, often, compromising vital societal functions (Pescaroli and Alexander, 2015; Menoni et al., 2017; Dominguez et al., 2021; Scaini et al., 2014). Comprehensive Post-Event Impact Assessments (PEIA) are necessary to qualify and quantify these cascading patterns through detailed analysis of the causality relationships between observed hazards and impacts during disasters.

60

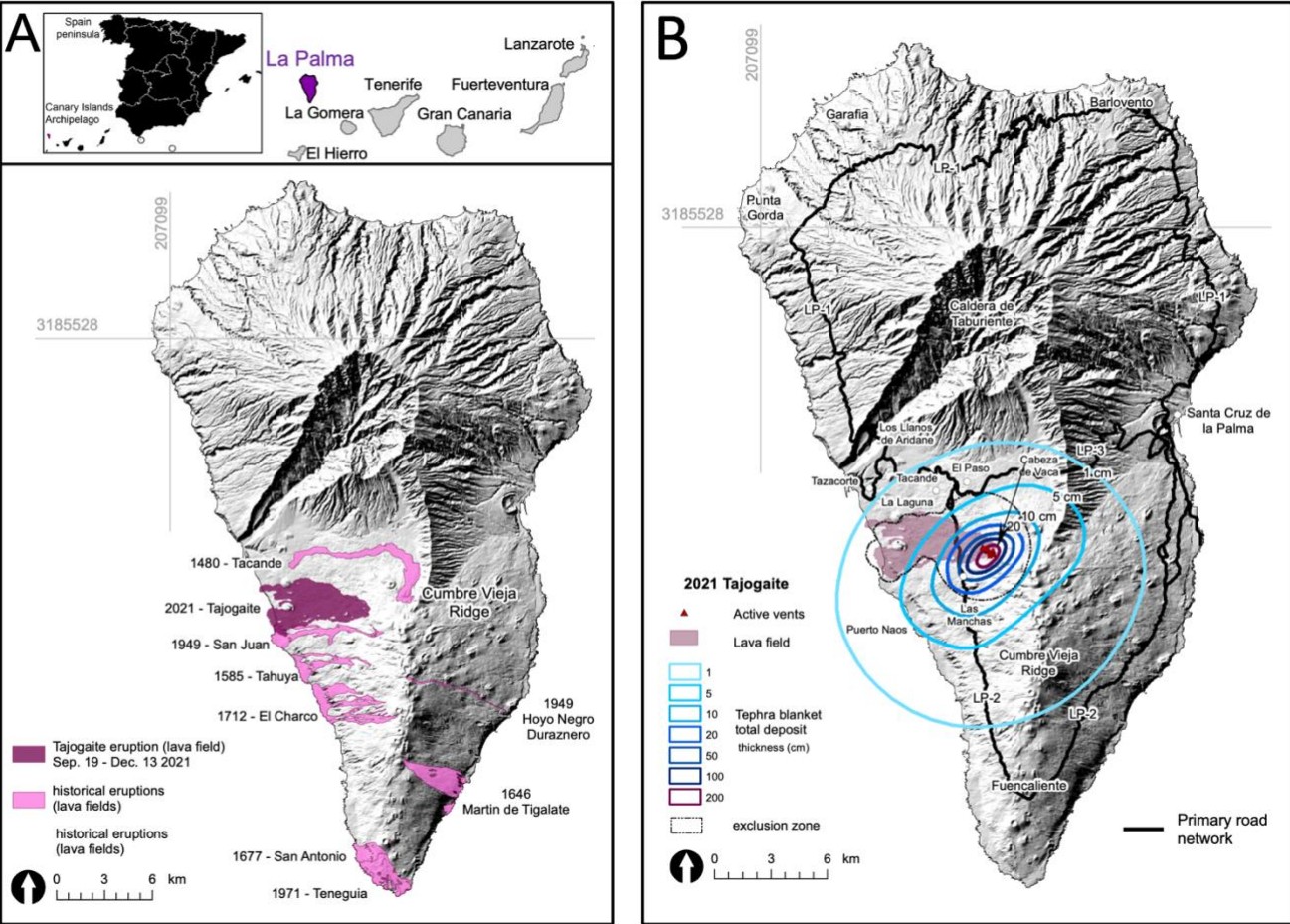

Figure 1. A) Context of La Palma Island relief showing the concentration of recent volcanism in the Southern Cumbre Vieja Ridge. At least 8 historical eruptions are recognized, although the evidence of hundreds of volcanoes is evidenced by the number of craters and cones revealed by the topography. Location of La Palma in the Canary archipelago and geographical distribution respect to Africa and Spain peninsula is also shown. B) 2021-Tajogaite total deposit of lava and tephra fallout (at the end of the eruption, Dec. 13, 2021) with the main localities affected. Primary roads are shown as a reference (black line). Data source: Lava footprint taken from Copernicus Emergency Management Service (© 2021 European Union, EMSR546). Tephra isopachs from (Bonadonna et al., 2022). Road network from © OpenStreetMap contributors 2021. Distributed under the Open Data Commons Open Database License (ODbL) v1.0. Hillshade from ESRI. All maps using the Projected Universal Transverse Mercator (UTM) zone 28 on the World Geodetic System WGS84 datum (EPSG:32628).

The 2021 Tajogaite eruption of La Palma Island (Spain) demonstrated how multi-hazard monogenetic volcanism triggered cascading effects to CI, resulting in a substantial disturbance in an insular environment. Recent volcanism (< 0.13 Ma) in La Palma is monogenetic, characterized by an extensive volcanic field comprising 10s – 100s of small cinder cones, each of those

being the result of single eruptive episodes (Fig. 1A). In such fields, forecasting the location and magnitude of future events is fraught with large uncertainties (Martí et al., 2016; Marrero et al., 2019; Martí et al., 2022), particularly due to rapid magma ascent (from 10 to 0 km depth in <7 days in the case of Tajogaite), and the possibility of lateral shallow magma migration along hydrothermally altered plumbing systems (observed at < 3 km depth within the few hours before the eruption began in Tajogaite, D'Auria et al., (2022)). These spatial and temporal uncertainties are especially challenging for crisis management of islands as an eruption location deviation of a few kilometres can entirely alter the hazard extent and consequently the response actions. Moreover the 85-day Tajogaite eruption produced multiple eruptive styles (i.e., Hawaiian, Strombolian, violent Strombolian; Bonadonna et al., 2023; Taddeucci et al., 2023) and multiple simultaneous products (i.e., gas, lava, tephra, Fig. 1B) from eleven vents (Romero et al., 2022; Civico et al., 2022). The interaction between this complex hazardous landscape and exposed human and environmental systems substantially compromised all social and economic sectors on the island (Rey et al., 2023; Troll et al., 2023). The first lava flows inundated the west coast and rapidly cut the primary road LP-2, the main longitudinal axis located at ~1.7 km W of the vents. A few hours after the eruption began, this lava flow effectively split the island in two, inducing a long-term disconnection between northern and southern communities and sectors (Fig. 1B). Due to the low redundancy of the road network caused by topographic constraints, major disruption of the ground transportation and water networks occurred across 10 of the 14 municipalities of the island. Cascading effects were subsequently triggered due to the interconnectivity of roads with societal functions such as emergency services and critical facilities (e.g., civil protection, schools), and economic sectors (e.g., agriculture, tourism). This case study highlights that the potential impacts for insular volcanism involving rapid eruptive changes and compound hazards over long durations requires comprehensive PEIAs capable of depicting the diverse and multi-dimensional consequences of affected societies (e.g., physical, functional, systemic, social).

Traditional PEIA focus on CI ranges from narrative descriptions to semi-quantitative studies that apply Impact States (IS) and/or Damage and Disruption States (DDS) generally associated to a physical Hazard Intensity Metrics (HIM). Wilson et al., (2014) proposed a conceptual model with an impact scale evolving as a function of increasing HIM (e.g., tephra thickness), ranging from tolerance (systems fully functioning) to disruption (operative at reduced functions until restoration) to damage (not functioning until repair). Similarly, Jenkins et al., (2015b) categorised five DDS levels as a function of tephra thickness, ranging from no damage (D0) to beyond economic repair (D5) for a range of CI systems. However, for CI networks, the role of systemic vulnerability (linkages including dependencies and connectivity considering modern societies as a system-of-systems) and response capacity of the relevant agencies is highly influential on the total impacts observed during disasters (GAR, 2024). These influences often manifest through complex causal relationships, rather than linearly correlating with physical HIMs, and therefore cannot be adequately described using existing damage and impact scales. Following the Dominguez et al. (2021) forensic framework, we apply here a PEIA to describe and quantify the cascading consequences of the 2021 Tajogaite eruption by integrating the physical damage, functional disruption and systemic orders of impact of CI.

A substantial knowledge gap in PEIAs is the quantification of CI network properties (e.g., connectivity, interdependence) and understanding how these properties evolve and provoke impacts during volcanic eruptions. To build on this PEIA, we explore the utility of graph network analysis as a powerful tool to assess the connectivity evolution of the road network over a volcanic crisis. Critical infrastructure networks can be represented and analysed using graph theory, a mathematical expression of relationships between vertices (nodes, e.g., electricity substations) connected by segments (edges, e.g., electricity lines) (Barthélemy, 2011). Pioneering studies that apply network analysis in volcanology use GIS-based constructions of graphs to quantify systemic vulnerability (e.g., connectivity, dependency, Weir et al., (2024b, a)), or complex interactions between networks and population demographics in evacuation (e.g., Mossoux et al., 2019). Specifically for tephra deposition, major efforts have been made to quantify the functionality of transport systems, expressed through skid resistance and visibility reduction (Blake et al., 2017b, a, 2018), road criticality metrics (Hayes et al., 2022), road network interdependencies (Weir et al., 2024b), and evacuation modelling (Wild et al., 2021). In the case of lava flows, impacts are often considered binary (presence/absence model of impact e.g., Meredith et al., 2023). However, crisis management and recovery can be strongly inhibited when CI are in close proximity to lava flows, though not physically impacted (e.g., Kim et al., 2019; Jenkins et al., 2017). The temporal evolution of graph indicators allows for an objective quantification of the road network connectivity, highlighting the importance of the overall disruption of the island directly or indirectly affected by compounded lava and tephra.

A general overview of the impacts to La Palma island during the 2021 Tajogaite eruption crisis has been presented by Rey et al. (2023) and Troll et al. (2023), and comprehensive reconstructions of impacts to the built-environment have been presented by Meredith et al. (2023), Reyes-Hardy et al. (2024), and Biass et al. (2024). In this study we present first a reconstruction of the cascading impacts across the three orders of impact (i.e., physical, functional and systemic). Secondly, we present a comprehensive network analysis of the loss of functionality of the road network, further quantifying the loss of connectivity of the island syn-eruption (lasting 3 months) and during the post-eruption recovery phase (6 and 18 months after the end of the eruption).

## 2 Chronology of the 2021 Tajogaite eruption (La Palma, Spain)

### 2.1. Eruptive history

Located in the Atlantic Ocean, 100 km off from the African mainland and 1400 km south of the Spanish peninsula, La Palma is one of the most active volcanic islands of the Canary archipelago (Carracedo et al., 2001). Over the past 150,000 years, activity is exclusively restricted to the monogenetic rift of Cumbre Vieja (Fig. 1A) with 8 of the 14 known eruptions in the Canaries since the Spanish conquest in 1493 (Supplementary Table S1). All these eruptions are associated with multiple eruptive vents along fissures, with both effusive) and explosive activity (Carracedo and Troll, 2016; Carracedo et al., 2022a). Following a precursory seismic period, which started in 2017, a new fissure opened on the western flank of Cumbre Vieja,

building a 187 m-high of a basaltic cone in 85 days (Civico et al. 2022), named Tajogaite or *montaña rajada* in the indigenous guanche language (Ecoavant, 2022). As a result of a synchronous effusive and explosive activity, Tajogaite produced compound lava, tephra and gas emissions. Over the eruption >9,000 earthquakes were recorded, and 2 Tg of $SO_2$ (in total) and up to 4,435 tonnes/day of $CO_2$ were released into the atmosphere (PEVOLCA report 25/12/21). Due to the complex, long-lasting consequences of gas emissions in La Palma (syn- and post-eruption, still ongoing in 2024), we focus only on impacts of lava and tephra. A total volume of ~23 x $10^7$ m$^3$, constituted by ~74% of lava flows, ~16% the scoria cone, and ~10% the tephra blanket, were emitted (Bonadonna et al., 2022). Approximately 190 km$^2$ and 12 km$^2$ were covered by tephra and lava, respectively; with ballistics reaching a radial travel distance of ~1.5 km (Figs. 1B, 2). Complex explosive dynamics combined with variable wind patterns resulted in a tephra blanket deposit of 3 units (lower -LU-, middle -MU-, upper -UU-) and 11 layers, mainly dispersed NE-SW (Figs. 1B, 2). The entrance of lava flows into the sea and major changes of lava geometry (width and length, Supplementary Fig. S1), as well as the temporal evolution on eruptive styles and wind patterns delimiting the different tephra units, had significant implications on the systems impacted and are relevant for the forensic analysis presented here.

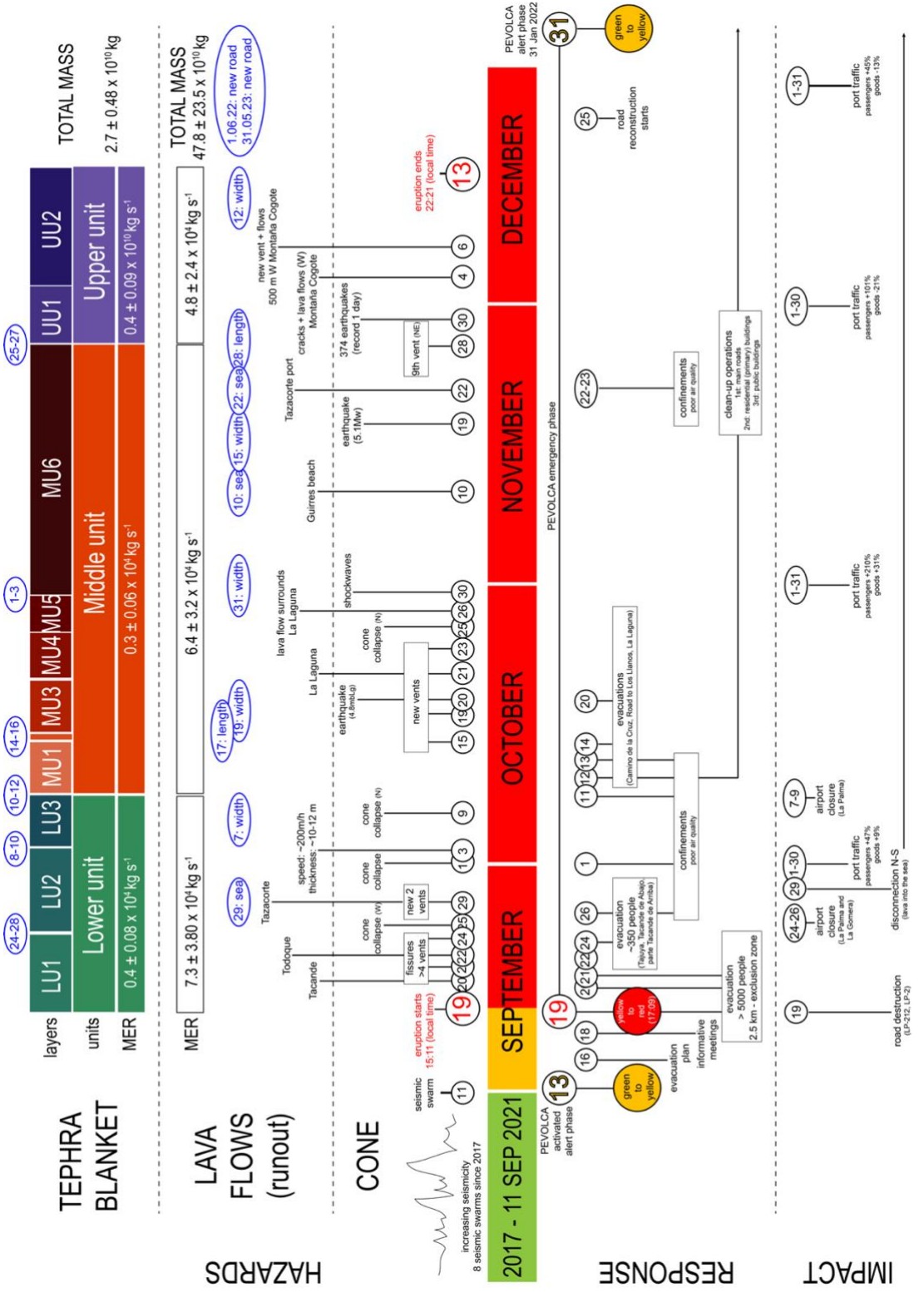

Figure 2. Chronology of the eruption including main events associated with hazards, response and impact on the transportation system, target of this study. Green, yellow and red colours indicate the change on the PEVOLCA volcanic level. Analysed hazards are tephra fallout blanket (distinct from the cone formation) and lava flows. MER corresponds to Mass Eruption Rates from Bonadonna et al., 2022. Black circles indicate dates of main hazard events. Blue ellipses indicate key inflection dates on the temporal evolution of these hazards; length and width indicate major changes on the geometry of the lava field and sea when a main lava channel reached the sea coast.

## 2.2. Volcanic crisis context

La Palma comprises 14 municipalities with a decentralised government that grants a high degree of autonomy in parallel with a strong hierarchical structure of five levels: Municipal, Insular (*Cabildo de La Palma*), Province (*Santa Cruz de Tenerife*), Community (*Comunidad Autónoma de Canarias*), and State (*Gobierno Nacional*). Most of the 83,380 inhabitants in January 2021 (Instituto Nacional de Estadística of Spain, 2021) are located in the dense urban area of the Valley of Los Llanos de Aridane where major commercial activities are developed (Fig. 3A). Local economy is primarily based on agriculture, followed by tourism. Crops of bananas, and secondarily grapes, avocados and citrus fruits, are produced across the island, but commercial packing and distribution as well as many of tourism sites are located in Los Llanos de Aridane (from now on, Los Llanos) (Fig. 3A). The characteristic topography of La Palma restrains land-use, urban development and infrastructure configuration (Fig. 3B).

The 2021 Tajogaite eruption led to the deployment of the volcanic emergency plan, attached to the Community government, and last updated in July 2018 (PEVOLCA, Plan Especial de Protección Civil y Atención de Emergencias por riesgo volcánico en la Comunidad Autónoma de Canarias, https://www.gobiernodecanarias.org/infovolcanlapalma/pevolca/). Following a seismic swarm with more than 400 earthquakes since Sep. 12, 2021, and confirmed magmatic activity (Helium-3 gas monitoring), the PEVOLCA plan was activated and the volcanic alert, based on 4 levels (Supplementary Fig. S2), was raised from green to yellow on Sep. 13, 2021 (PEVOLCA, report 13/09/21) (Fig. 2). Rapid magma ascent and shallow migration revealed by seismic tomography occurred during the seven days prior to the eruption (D'Auria et al., 2022), initially suggesting the vent would emerge ~6 km SE of the observed first vent opening in Cabeza de Vaca (Fig. 1B). In the absence of an acceptable level of confidence in the vent opening location, those with reduced mobility in the 4 forecast affected municipalities (Los Llanos, El Paso, Tazacorte and Fuencaliente; Figs 1B, 3A) were preventively evacuated in the morning of Sep. 19 (Brusini Dominguez, 2022). The eruption began in Cabeza de Vaca that same day at 15:11:00 (all times reported as local). The alert level was then updated from yellow to red at 17:09:00, and an exclusion zone of 2.5 km radius around the opened vents was shortly established (PEVOLCA report 19/19/21 evening) and the General Directorate of Civil Protection and Emergencies of Spain requested the rapid mapping of the Copernicus Emergency Management Service (© 2021 European Union, EMSR546, hereafter referred to as Copernicus) to facilitate local emergency measures and communication.

Although the focus was on the evolution of the lava flows, the simultaneous production of tephra and gases affected also other islands (i.e., La Gomera, El Hierro, Tenerife and Gran Canaria) and the accumulation of ~190 km$^2$ of tephra deposit exacerbated emergency and daily activities in La Palma. The conjunction of all these volcanic products generated ~€842M of direct losses (€228M in road damage, €200M in crop damage and associated loss of production, and €165M in building damage) and substantial uncalculated indirect losses (BOE, 2022).

### 2.2.1 Response

While all decisions before and after the emergency phase are the responsibility of each Municipality and Cabildo, during the emergency phase, all decisions were managed by the PEVOLCA Advanced Command Post advised by the Emergency and Scientific Committees (PEVOLCA report 25/12/21). PEVOLCA structure is constituted by six groups (i.e., Intervention, Health Care, Security, Technical Support, Logistics and Essential Services) whose principal aim is preserving life during volcanic crisis. Here we describe 5 main response measures, amongst many others, that were fundamental to mitigate direct fatalities, reduce impact and maintain a minimum level of continuity for affected communities (Fig. 2). First, rapid evacuation of 7,000 people was carried out in 8 phases, where people were housed in hotels, with relatives and at emergency shelters (Brusini Dominguez, 2022). Second, clean-up operations of tephra for main roads as well as primary residential and public buildings (e.g., schools and health centres) were managed by the *Unidad Militar de Emergencias* (UME) from mid-October to eruption cessation. Third, temporary access was allowed to residential property in the exclusion zone to collect building contents and safeguard property (e.g., through cleaning roofs of accumulated tephra). This unprecedented decision in Spanish emergency management history required the mobilization of all emergency management stakeholders (i.e., civil protection, fire brigades, CECOPIN (*Centro de Coordinación Operativa Insular*), police and civil guard agents). Fourth, the coordination of all the six intervention groups of PEVOLCA ensured the continuity of critical services supply, such as electricity, water and gas. Fifth, the support of the Municipal and Insular governments (*Cabildo*), together with the intervention groups, facilitated daily activities to avoid the total economy and social paralysis. These actions included the efforts taken to reconnect crop production sites located south of the lava flows with market distribution located in Los Llanos in the north, thanks to the transport of workers by boats or buses. There were also significant efforts to relocate evacuated children and facilitate the transport to the schools that were not physically impacted across the island, promoting parents to continue daily activities.

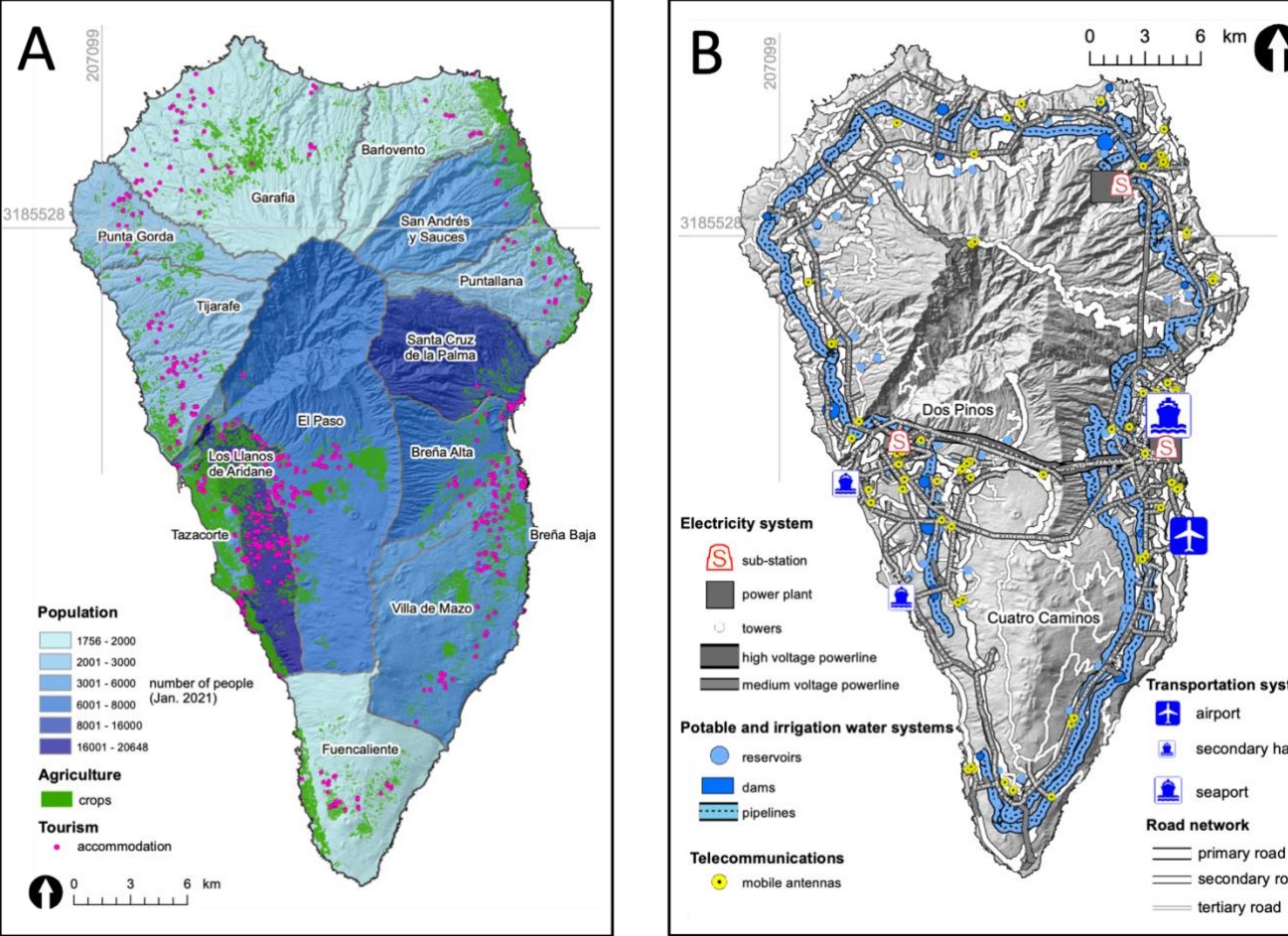

Figure 3. A) La Palma population density distributed by municipality. Main economic sectors, agriculture (represented by crops) and tourism (represented by accommodation sites) are also shown. B) Critical infrastructures including electricity, potable and irrigation water, telecommunications and transportation systems. Data source: Road network from © OpenStreetMap contributors 2021. Distributed under the Open Data Commons Open Database License (ODbL) v1.0. Economic and infrastructure layers available in La Palma Open Data repository (Cabildo Insular de La Palma) and https://www.lapalmaaguas.com/visor/. Population density from Instituto Nacional de Estadística, INE, 2021. Hillshade from ESRI. All maps using the Projected Universal Transverse Mercator (UTM) zone 28 on the World Geodetic System WGS84 datum (EPSG:32628).

## 3. Methods

This PEIA consist of two main tools, i) a forensic approach to describe and structure the cascading impacts triggered by the road network disruption, and ii) a comprehensive network analysis to quantify the loss of connectivity of the road network syn-eruption (lasting 3 months) and during the post-eruption recovery phase (6 and 18 months after the end of the eruption,

June 30, 2022 and June 30, 2023, respectively). Following the conceptual framework proposed by Dominguez et al. (2021), we applied the 4-steps PEIA shown in Figure 4 and described below.

Figure 4. Post-Event Impact Assessment conceptual framework. Four-steps methodology adapted from Dominguez et al., (2021) and complemented with network analysis to quantify the loss of connectivity of the road network and subsequent consequences in travel time. In green field campaigns dates.

## 3.1 Data collection

Impact data was collated from field observations, literature review, and discussions with stakeholders during three main campaigns, in Oct. 21-Nov. 5, 2021 (syn-eruption), and in Feb. 7-18, 2022 and May 14-20, 2022 (post-eruption). Following the guidance of the Science-Practice-Policy Interface, we apply here a stakeholder engagement model in the basis of an operating environment (Tambe et al., 2023; Wyborn et al., 2017). Since this research was conducted during and immediately

after the eruption, we adopt ad hoc interactions with stakeholders that were highly time-limited and operationally busy during the emergency management, response and recovery periods. Interactions were therefore semi-structured, with several recurring discussion points. This approach ensured that interactions were mutually beneficial, effective and highly adaptive. Depending on availability, discussions included one-to-one interactions or grouped stakeholders from Institutions (e.g., Government, Civil Protection) and CI managers and technicians (e.g., roads, electricity, water) (Supplementary Table S2). A complete set of questions, adapted from Dominguez et al., (2021) and focused on specific sectors and systems of La Palma helped to support this forensic PEIA to explore the "what, when, how and why" impacts occurred (Supplementary Table S3). A first analysis of all sectors and systems affected (Steps I and II, Fig. 4), showed that the major disturbance on the island was due to the physical damage of the primary road LP-2, which resulted in a substantial cascading chain of impacts to emergency and critical facilities as well as economic sectors, especially agriculture. Spatiotemporal evolution of both lava and tephra fallout blanket was crucial to understand the direct and indirect interaction with the systems of La Palma. Lava flow field evolution was estimated from Copernicus, which provided georeferenced lava footprints for 55 of the 85 days of the eruption. For the same dates, tephra footprints were retrieved from the tephra deposit reconstruction, based on isopach data (Biass et al., 2024; Bonadonna et al., 2022). Additional key dates of tephra sedimentation for different layers and MER major changes for both tephra and lava were provided by Bonadonna et al., (2022, 2023) (Fig. 2).

### 3.2. Data processing and analysis

### 3.2.1. Cascading impacts

Data from the interactions with stakeholders and literature (Steps I and II, Fig. 4) fed the root cause and consequence analysis necessary to structure and map the cascading impacts due to lava and tephra on the road network (Step III, Fig. 4). The sequence of cascading impacts was established using deductive reasoning, exploring the root causes of physical damage ([1st] order impact), loss of functionality of the road network ([2nd] order impact) and the subsequent consequences to other sectors ([3rd] order impact). Based on a series of logical questions we identify five levels of road functionality reduction, from no loss of functionality (FL-0) to permanent closure (FL-IV), defined in Table 1 and detailed in the Results section. Reduced service due to the direct impact of tephra accumulation (FL-II, Table 1) was further sub-divided into five levels of disruption, applying Impact States (IS) as a function of tephra thickness, previously defined by Blake et al., (2017a) (Table 2). The three orders of impact are then mapped based on the qualitative descriptions made by the stakeholders combined with the spatiotemporal evolution of hazards. Quantification of affected road lengths and crops surfaces was determined through GIS analysis.

Table 1. Functionality levels of roads as a consequence of the [2nd] order of impact: loss of functionality of the road network. Impact States defined by Blake et al. (2017a) are applied here to subdivide the Functionality Level – II. These Impact States were associated with 3 speed restriction scenarios, shown in Table 2.

| Level of functionality | Description |
|---|---|
| FL - 0 | No loss of functionality |
| FL – I | Disrupted service -> indirect impact due to transferred functions (e.g., increase traffic) |
| FL – II | Reduced service -> direct impact due to tephra accumulation. Sub-levels of functionality based on Impacts States as a function of tephra thickness are shown in Table 2 |
| FL – III | Reduced service -> indirect impact due to exclusion restrictions (i.e., 2.5 km buffer of lava flows) |
| FL – IV | Permanent closure -> direct impact due to presence of lava flows |

**Road network analysis**

Road network analysis was performed using OpenStreetMap (OSM) database to build a connected graph of nodes and edges (Boeing, 2017). To quantify the loss of connectivity of the road network due to lava and tephra, we intersected the OSM road network previous to the eruption (Sep. 18, 2021) with the snapshots of lava footprints at the available 55 dates throughout the eruption, and 2 dates post-eruption (June 30, 2022, June 30, 2023) to obtain the spatial change of the road network across time (58 dates in total). To explore the contribution of hazard effects, graph centrality indicators were calculated considering the impact of i) only lava, and ii) compounded lava and tephra fallout (hereafter referred to as lava+tephra). In the case of tephra, we retrieved the thickness to infer the corresponding impact level based on the IS of Blake et al. (2017a) and we applied a reduction in travel speed as a function of tephra thickness on the road edges (Table 2). Speed restrictions result in an increasing travel time, an indirect measure of the loss of functionality of roads. In the absence of literature on travel speed reduction related to tephra fallout on roads, we test three speed reduction scenarios as a function of IS that include low (LS) and high (HS) speeds, computing a reduced fraction of the original Velocity (V) per road type (e.g., primary, secondary) (Table 2). A third speed scenario, Recommended Scenario (RS), was inferred from the generic recommendations by the Spanish traffic agency of Spain (*Dirección General de Tráfico, DGT)* under adverse weather conditions (i.e., rain, snow, fog). These recommendations are based in the Spanish 4-disruption levels scale, codified by colours, and associated speed reduction that has been adapted to the IS due to tephra (Table 2).

Table 2. Correlation of tephra thickness with Impact States (IS) of Blake et al. (2017a) and speed restriction scenarios. V, Velocity, corresponds to the original speed for each road type in La Palma. Recommended speeds come from the Dirección General de Tráfico (DGT) of Spain for 5 levels of functionality under adverse weather conditions (e.g., rain, wind) available at https://www.dgt.es/comunicacion/noticias/planificar-el-viaje-con-la-mejor-informacion/. Notice that for IS3, although considered null, a value of 1km/h has been used for mathematical calculations.

| Tephra thickness (cm) | IS | IS description | Low Speed (LS) | High Speed (HS) | Recommended Speed (RS) | DGT levels of functionality |
|---|---|---|---|---|---|---|
| 0 – 0.01 | IS0 | No disruption | V * 1 | V * 1 | V | No restriction |

| >0.01 – 0.1 | IS1a | Minor skid resistance, reduction possible | V * 0.5 | V * 0.75 | V (80 km/h max) | Conditioned circulation |
|---|---|---|---|---|---|---|
| >0.1 – 0.5 | IS1b | Skid resistance reduction likely | V * 0.25 | V * 0.5 | V (60 km/h max) | Irregular circulation |
| >0.5 – 10 | IS1a | Minor skid resistance, reduction possible | V * 0.5 | V * 0.75 | V (80 km/h max) | Conditioned circulation |
| >10 – 25 | IS2 | Impassable for some vehicles | V * 0.1 | V * 0.25 | 30 km/h | Difficult circulation |
| > 25 | IS3 | Impassable if tephra unconsolidated | 1 km/h | 1 km/h | 1 km/h | Interrupted circulation |

The connectivity of the island can be estimated in terms of centrality indicators (Table 3) (Boeing, 2018). Here, we focus only in two indicators calculated for edges, *betweenness centrality (EBC),* which indicates how often an edge is utilised in the shortest path calculation between nodes, and *closeness centrality (ECC)*, which indicates the relative closeness of nodes/edges to others by summing the edge length (if representing a spatial referenced graph). Since *OSMnx* allows for betweenness centrality to be calculated for weighted edges (i.e., calculation based on travel time, speed or length, Table 3), we apply the travel speed reduction scenarios (LS, HS, RS) to modulate the travel time (EBC-travel time) and/or length (EBC-length). This allows the analysis of speed reduction due to loss of visibility and skid resistance associated with tephra. In order to compare and analyse the evolution of indicators across time, the median EBC and ECC for the whole island network has been calculated for each date.

Table 3. Summary of network parameters used in this study. Definitions from Boeing (2017).

| Parameters | Definition |
|---|---|
| **n** | Number of nodes |
| **m** | Number of edges |
| **Average node degree (k avg)** | Average number of inbound and outbound edges incident to the nodes |
| **Intersection counts** | Number of intersections in the network |
| **Street length total (m)** | Sum of edge lengths corresponding to one street |
| **Average circuity (c avg)** | Average ratio between edge length and the straight-line distance between adjacent nodes |
| **Betweenness centrality (BC)** | Fraction of all shortest paths that pass through each node (NBC) or each edge (EBC). Calculations based on 3 weights: travel speed, length and travel time |
| **Closeness centrality (CC)** | Reciprocal of the sum of the distance from the node (NBC) or edge (ECC) to all other nodes/edges in the graph, weighted by length. The more central a node is, the closer to all other nodes |

Finally, the impact of loss of functionality of roads on other systems during the eruption is calculated using the change in travel time for selected routes. The Origin-Destination (O-to-D) points correspond to locations that stakeholders recurrently mentioned during discussions. Travel time for four routes corresponding to target sectors (i.e., emergency, health, agriculture and education; Table 4) have been analysed.

Table 4. Origin-Destination points for key target systems. The football stadium and Plaza Bonita correspond to evacuation meeting points decreed by PEVOLCA, few days before the eruption (Brusini Dominguez, 2022).

| Route | Target system | Origin Point | | | Destination Point | | |
|---|---|---|---|---|---|---|---|
| | | Name | UTM - X | UTM - Y | Name | UTM - X | UTM - Y |
| 1 | Emergency | Football field (Las Manchas) | 218336.94 | 3167571.22 | CECOPIN | 228221.22 | 3173662.91 |
| 2 | Health | Plaza Bonita (Las Manchas) | 217463.83 | 3166548.29 | Hospital | 227302.06 | 3174895.91 |
| 3 | Agriculture | Plaza Bonita (Las Manchas) | 217463.83 | 3166548.29 | Los Llanos municipality | 217489.71 | 3166500.33 |
| 4 | Education | Fuencaliente downtown | 221479.84 | 3155301.73 | Secondary school | 218150.95 | 3172304.15 |

### 3.3. Validation

Finally, we identify the critical aspects that aggravated cascading consequences, by discretising the causal order of impacts and quantifying the reduced functionality of the road network due to increasing travel times (Step IV, Fig. 4). An incremental validation of our results based on field observations, specifically on the travel time of the four routes and general connectivity of the island, was made two and three years after the eruption (May 2023 and 2024). We also explored the applicability of this methodology by analysing what would be the effect of historical lava footprints on the existing road network connectivity.

### 3.4. GIS setup and data source

Road network analysis was performed using the OSMnx python package 1.9.3 that downloads and converts OSM to a graph with corrected and simplified topology (Boeing, 2017). Road network data source from © OpenStreetMap contributors 2021, 2022 and 2023, distributed under the Open Data Commons Open Database License (ODbL) v1.0. Centrality indicators (Table 3) were estimated using the *NetworkX* Python package (Hagberg et al., 2008). Critical facilities, crops and tourism were retrieved from La Palma Open Data repository (https://www.opendatalapalma.es/). GIS setup was based on Python v3.9 and Geopandas 0.14.0 library with the Projected Universal Transverse Mercator (UTM) Coordinate System, zone 28N on the World Geodetic System WGS84 datum (EPSG:32628). Post-processing and map layouts were conducted in ArcMap v-10.7.1.

## 4. Results

### 4.1. Impacts to critical infrastructure

From the onset of the eruption the operational priorities of CI systems were to support emergency activities (e.g., coordination of restricted areas, clean-up operations) and to maintain a basic level of service for vital functions (i.e., shelter, water, sanitation, power, access) in the affected area and throughout La Palma. Impacts on the transportation system and cascading effects on the water system and agriculture are presented in Fig. 5 and described below. Details on other sectors are available in the Supplementary material.


- Ground transportation (roads)

Ground transportation, along with parallel electricity and water lines, suffers from low redundancy, as these networks obey a topographically-limited geometry where a primary ring network is fed by secondary and tertiary urban roads concentrated in the main populated areas (Fig. 3B) (a ring topology refers to a node connected to other two nodes by a single pathway as in a
ring shape, Dekker and Colbert, 2004). The north-south corridor on the western flank of Cumbre Vieja was almost immediately affected a few minutes after the eruption started, with the lava flow burying the secondary road LP-212 (Fig. 5A) and the primary road LP-2 (Fig. 5C). All roads within a 2.5 km radius of the vent and at the front of the lava flows were closed (exclusion zone). Given the high uncertainty of evolving lava paths, real-time monitoring by Civil Guards was conducted to aid emergency coordination, in parallel with tephra clean-up operations to clear evacuation routes. Major concerns about the
speed of lava flows and the time it would take to reach the Atlantic Ocean made local and global newspaper headlines. Emergency services anticipated large explosions and high gas release to occur within the initial days of the eruption, but it was not until the night of 28-29 September that first lava flows reached the sea (Fig. 5B). From then on, the island was severed in two, generating major disruptions to the road network, emergency services and the daily activities of La Palma.

- Maritime transportation (ports)

La Palma is connected to the Canary Archipelago by maritime and air routes, with maritime infrastructure the most crucial for its subsistence. The first emergency response actions were the delimitation and adjustment of the aerial and maritime exclusion zones in coordination with the Maritime Captaincy and Air Safety agency. Most goods, particularly oil, fuel, and mechanical products, come from the Spanish peninsula or Tenerife through the single seaport located in Santa Cruz de La Palma (Fig. 3B).
International and mainland visitors access the island by plane whilst local tourists and residents generally come by boat or ferry via Tenerife or La Gomera. Although some ash fell in the port for a few days of the eruption, there was no damage of port infrastructure or ships reported. Instead, the port served as a substitute for airport functions during its closure (Fig. 2). The major indirect impact to this sector was the significant increase of passenger traffic (an increase of 101% on average throughout the eruption, with a peak increase of 210% in October 2021 as compared to 2019) and a slight increase in the traffic of goods
(+1 % on average throughout the eruption, with a peak of +31% in October 2021 and low of -21% in November 2021 with

respect to 2019) (Fig. 2). The increase in passengers is due to the movement of emergency and scientific staff, but also with the so-called "volcano tourism", which substantially amplified the transient population in La Palma during the eruption (Dóniz-Páez et al., 2023). Additionally, a secondary small harbour for fishing and touristic activities located in Tazacorte (Fig. 3B) became important during the eruption to temporarily reconnect the north and south of the lava field for important rehabilitation

works, particularly for the water pumping system (e.g., UME water tanker, transportation of crop workers, Fig. 5G). Depending on the weather and eruptive conditions, these boats took ~30 minutes to cross a relatively short distance (about 3-5 nautical miles).

- Potable and irrigation water

The water supply system is intrinsically related to the island's geomorphology and topography. The major underground aquifers, located ~1000 m a.s.l in the Caldera de Taburiente, feed the galleries that supply 14 reservoirs configured along the N-S flow corridor (Fig. 3B). The closest reservoirs to the lava field are Dos Pinos in the north (384,000 m$^3$) and Cuatro Caminos in the south (108,000 m$^3$) (Fig. 3B). Although neither of these reservoirs were directly affected by the lava, the distribution lines (ground) that run parallel to the road network were buried, effectively dividing the water supply system in two (Fig 3B).

This necessitated the pumping of water uphill towards the summit of the Cumbre Vieja Ridge to divert supply to Cuatro Caminos for gravity-fed irrigation of banana crops in the south of the lava field (Fig. 3A,B). Agriculture, and in particular, banana crops, are highly dependent on the water irrigation system (Fig. 5H). La Palma produces ~65 Hm$^3$ of water per year whilst ~48 Hm$^3$ are dedicated to agriculture and ~9 Hm$^3$ to the population and tourism. Since direct impacts due to tephra fallout necessitated cleaning and irrigation of crops, three additional response measures where implemented: one desalination

plant located in Puerto Naos, producing 5,600 m$^3$ water per day, a water tanker ship managed by the UME provided 8,000 m$^3$ per day, and water pumping from Fuencaliente in the south. All these methods required substantial water and energy resourcing. A total of 3.7 km$^2$ of crops were buried by lava and 4.12 km$^2$ affected by tephra accumulation, and subsequent socio-economic consequences such as loss of livelihoods and psychosocial impacts were important. A severe systemic impact was associated with the disconnection of crops (in the south of the lava field, Fig. 5I) and the distribution/processing sites in

Los Llanos (in the north). Farm workers needed transportation from north-south every day via boat (Fig. 5G) or eastern road transport routes. Economic losses for the banana crop industry are estimated at €200M, calculated 6 months post-eruption, (BOE, 2022).

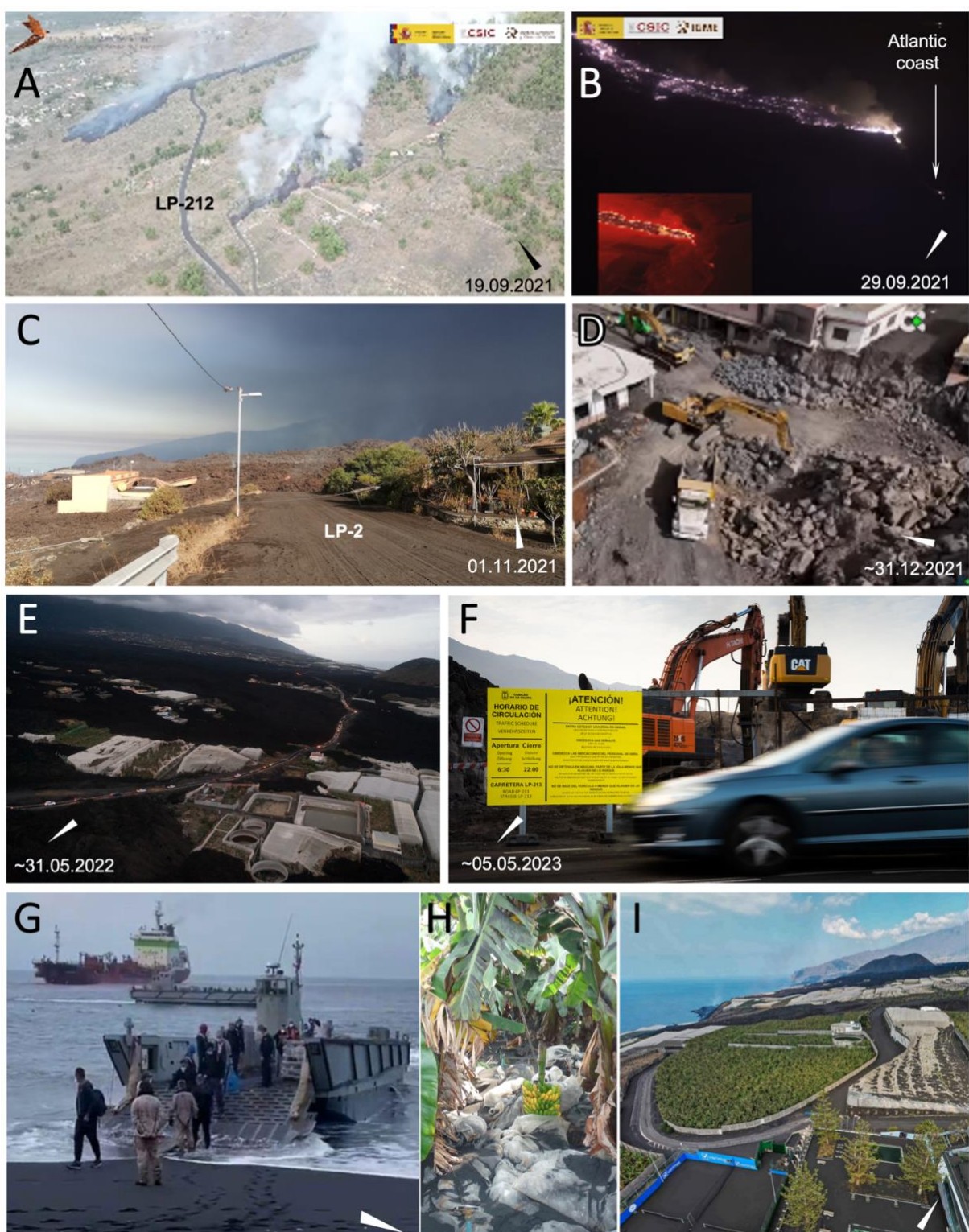

Figure 5. Impacts on critical infrastructure. Arrows indicate the North of the island. A) Secondary road LP-212 few minutes after the beginning of the eruption on September 19, 2021. UAV flight from IGME (2021a). B) Lava flows entering into the sea the night of September 29, 2021. UAV flight from IGME (2021b). C) Primary road LP-2 cut by lava flows and covered by tephra. Telecommunication wood pole damaged (Photo: L. Dominguez). D) Recovery works start by the end of the eruption (Television Canaria, 2023). E) New road opening on May 2022 (El Time news, 2022). F) Time and speed restrictions of the new road (Photo: L. Dominguez). G) Military boats transporting water and workers from Los Llanos to Puerto Naos (Planeta Canario news, 2021). H) Water irrigation needs in banana crops (Photo: L. Dominguez). I) Crops and water tanks South of lava field (Planeta Canario news, 2021).

## 4.2. Cascading impacts: mapping the causal order of impacts from the road network disruption

Due to its long duration and compound lava and tephra, combined with an insular low redundant environment, the 2021 Tajogaite eruption caused a 3-order of cascading impact starting with the physical damage of roads (1st order), the loss of functionality of the road network (2nd order), and multiple cascading effects on the water system, agriculture and all societal activities in the island (3rd order). The overall cascading effects of the 2021 Tajogaite eruption at various temporal and spatial scales are summarized in Table 5, while the overall cascading impact at the end of the eruption is shown in Fig. 6.

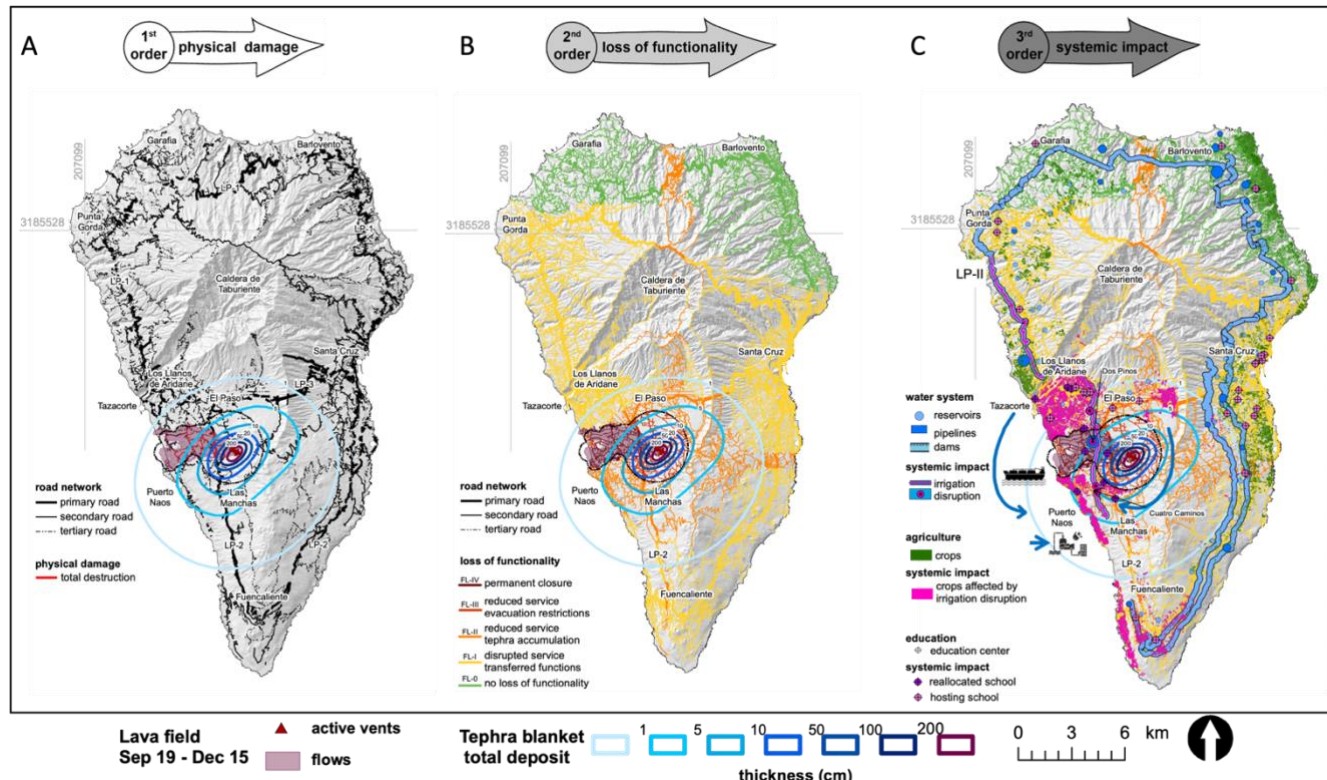

Figure 6. Spatial distribution of cascading impacts at the end of the eruption (Dec 13, 2021) described by A) physical damage of roads, B) loss of functionality of the road network, and C) systemic impacts on water, agriculture and education sectors due to the loss of connectivity of the island. Blue arrows in C) illustrate measures overtaken to transport water from north-to-south

to mitigate impact on agriculture. Data source: Lava footprint taken from Copernicus Emergency Management Service (© 2021 European Union, EMSR546). Tephra isopachs from (Bonadonna et al., 2022). Road network from © OpenStreetMap contributors 2021. Distributed under the Open Data Commons Open Database License (ODbL) v1.0. Economic and infrastructure layers available in La Palma Open Data repository (Cabildo Insular de La Palma) and https://www.lapalmaaguas.com/visor/. Hillshade from ESRI. All maps using the Projected Universal Transverse Mercator (UTM) zone 28 on the World Geodetic System WGS84 datum (EPSG:32628).

Table 5. Summary of impacts, response, temporal and spatial scales for the first, second and third order of impacts on the road network due to lava and tephra. Mun. indicates municipalities.

| | First order Physical damage | | Second order Loss of functionality | | Third order Systemic impact |
|---|---|---|---|---|---|
| | Lava | Tephra | Lava | Tephra | Lava and tephra |
| Impact | Total destruction | Total destruction on the cone formation area Pavement damaged due to clean-up operations | N-S disconnection in the western flank of Cumbre Vieja. Significant increase of travel time, particularly between Los Llanos and Las Manchas | Disruption due to reduced visibility, loss of traction, covering of marks, reduction of skid resistance, depending on deposit thickness | Difficult accessibility to repair all lifelines. Disruption of all daily activities due to road connectivity loss. Agriculture strongly affected (loss of water irrigation and workers accessibility) |
| Response | Permanent monitoring by mobile units of the Civil Guard to delimit exclusion and closure of roads | | Rapid recovery reconnects the network on May 2022 and Jun. 2023 | Clean-up prioritization: evacuation routes almost immediately. Secondly, accessibility to houses | Reconnection of E-W water sources and fast construction of desalination plants. Transport of workers and water with UME ship Reallocation of school students and implementation of bus transportation |
| Metrics | 73.8 km (12% of the whole network) from which primary, 2.3 km; secondary >5.8 km and >65 km of all other roads. | Considered null | Estimated scenarios and road lengths considering both lava and tephra are detailed in Figures 5B and 6. | | Estimated scenarios shown in Figure 5C. |
| Temporal scales | Permanent | Permanent (unless | Temporal months-years | Temporal days-weeks-months | Temporal – permanent |

| | | | | | |
|---|---|---|---|---|---|
| | pavement reparation) | (for the network, since buried roads are permanently damaged) | (permanent burying close to the formed cone) | months for transportation (UME ship left the island on Dec. 2021).<br><br>Some reallocations can be considered permanent | |
| Spatial scales | Local 3 mun.<br><br>Total surface: 12.19 km²<br><br>Gradual evolution as a function of emission rates and viscosity changes | Punctual sites | Regional 10 mun.<br><br>Total surface: ~12 km² | Insular – Regional La Palma – Canary archipelago<br><br>Deposit surface: ~190 km²<br><br>Gradual evolution as a function of eruptive styles and wind (other islands) | Insular – Regional – National* – International*<br><br>Specific cases such as migrations to other islands or peninsula occurred.<br><br>*Agriculture and tourism consequences affected Spain and even the European Community (agriculture subventions) |

### 4.2.1. First order impact: physical damage of roads

The total destruction of roads was mainly caused by i) the formation of the cinder cone, which buried the roads located in proximal area, and ii) lava flows which permanently buried a total of 73.8 km of roads (~12 % of the total La Palma network), from which 2.3 km were primary (LP-2), >5.8 km secondary (LP-212, LP-213, LP-211), and 0.3 km tertiary (LP-2132) roads, in addition to hundreds of urban streets (Fig. 6A, Table 5). Pavement of roads was also indirectly damaged by tephra clean-up machinery in some places, including cracks, deep scratches and marking abrasion.

### 4.2.2. Second order impact: loss of functionality of the road network

There is no exhaustive record of all disruptions that occurred. Indeed, the DGT of Spain recorded and reported only the largest disruptions to road network operations. To supplement this data gap, we attempt to reconstruct here a set of functionality scenarios based on the narrative descriptions of stakeholders and deductive logic questions presented in an event tree shown in Fig. 7A (Functionality Levels, FL 0 to IV, described in Table 1). The second order of impact for the final day of the eruption (Dec. 13, 2021) is shown in Fig. 6B whilst the temporal evolution of FL throughout the eruption is shown in Fig. 7B. Major disruption due to permanent closure of the road (FL-IV) was always associated with physical damage caused by lava flows, which increases from 0.5% of the total road network in the first 2 days, to 2.9% at the end of the eruption. In the exclusion zone, with a maximum of 6% of the total network, there was a substantial reduction (FL III) in road service, leading to considerably increased traffic and travel times, especially between Fuencaliente and the rest of the island. In addition,

management of the exclusion zone and temporary access of residents for collecting house contents and roof cleaning required substantial coordination efforts. Disruption due to tephra deposition was minor (FL-II), as principal emergency routes were frequently cleaned. However, as detailed data for these clean-up operations and response measures were not available, we assumed a scenario of total accumulation of tephra where no clean-up operations were implemented.


Approximately 16% of the total road network of the island was affected by > 0.1 mm of tephra by the end of the eruption, causing a FL-II level (Figs. 6B, 7B). This disruption level was then sub-divided in Impact States (IS) associated with tephra thickness thresholds (Table 2). Figure 7C shows the total road length in the different ISs throughout the eruption. In the early stages of the eruption most of roads reached IS1a and IS1b. Then, IS2 and IS3 monotonically increase with time, corresponding
to the growth of the scoria cone and sedimentation of the tephra blanket, particularly rapid during the deposition of the LU unit until Oct. 12 (Fig. 7C). The total road length in IS1b follows an interesting behaviour, being highest at the beginning of the eruption then decreasing to 0 towards the sedimentation of MU, where values suddenly increase and monotonically decrease to 0 on Nov. 1-3 (fig. 7C), corresponding to the MU5-MU6 deposition (Fig. 2). Approximately 3,000 km of roads across the island were not affected by tephra, shown in a steady IS0 throughout the eruption (Fig. 6C).


Finally, minor disturbances to the road network occurred in areas not directly impacted by lava and/or tephra. However, all stakeholders mentioned that traffic congestions were frequent between Punta Gorda in the west and Santa Cruz de La Palma in the east (FL-I; disrupted service, Fig. 6B), but no information is available to reconstruct the temporal evolution of this zone nor to discard the fact that other consequences could have occurred in the North of the island, for which, the level FL-0 is
assigned (Figs. 6B, 7A and 7B).

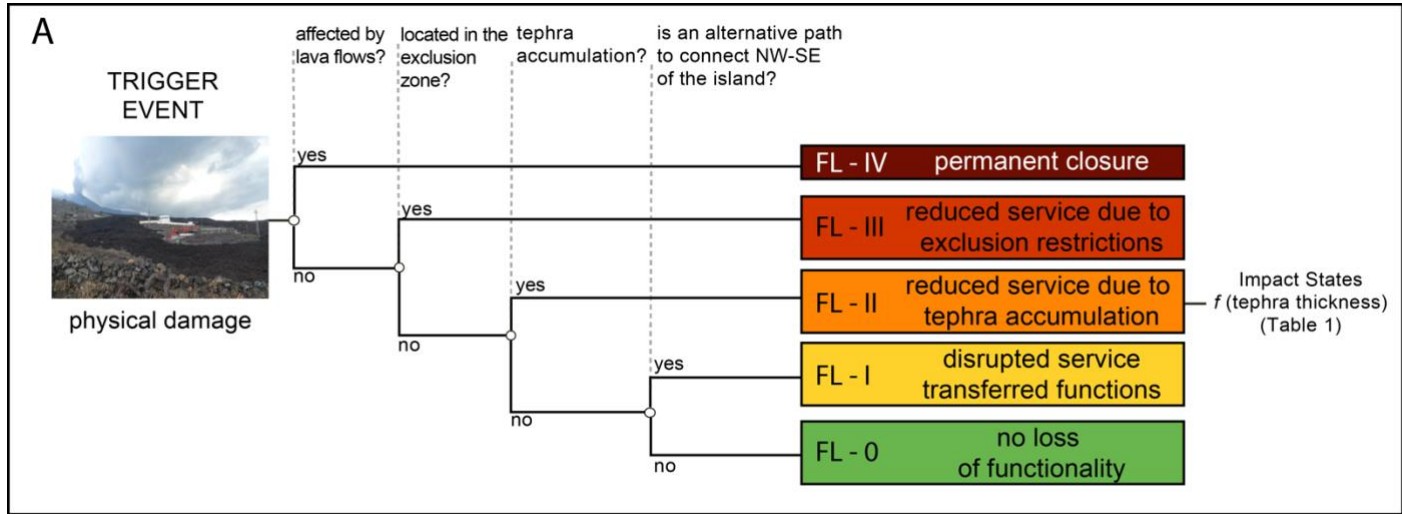

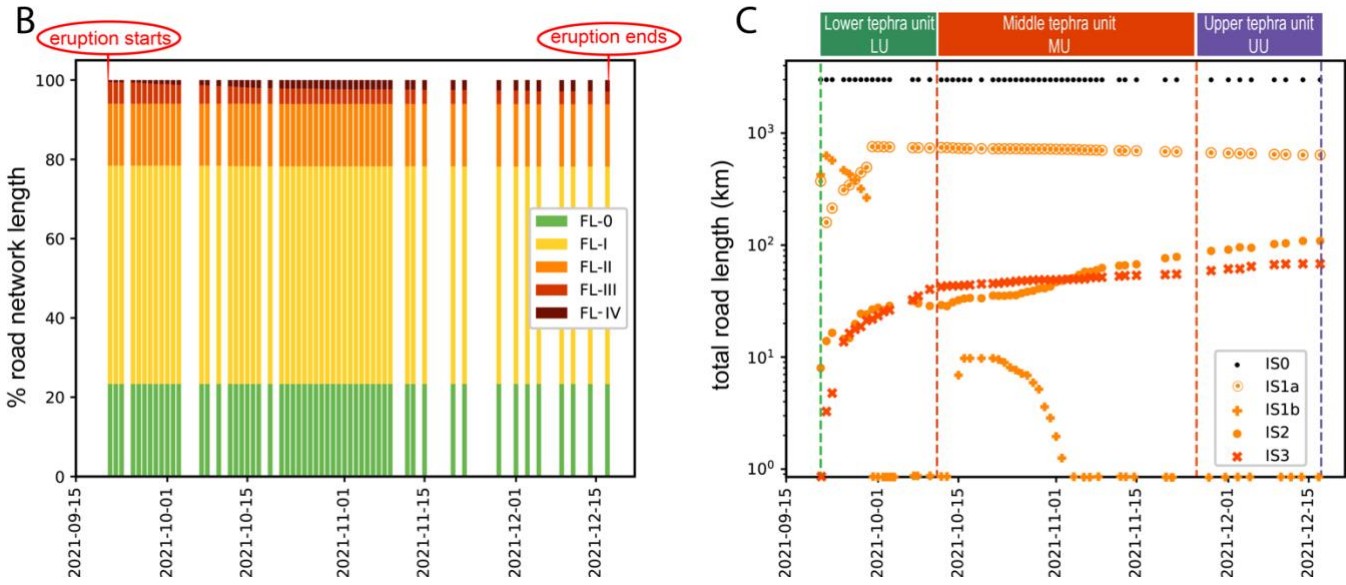

Figure 7. Second order of impact: loss of functionality of the road network. A) Event tree showing the 5 levels of functionality (FL) on the road network. B) Temporal evolution of Functionality Levels (FL) expressed in percentage of the total road network length. C) Temporal evolution of Impact States (IS), associated with tephra thickness thresholds of Table 2, expressed in total length of roads affected. Tephra deposit units are shown to indicate major changes on tephra sedimentation throughout the eruption.

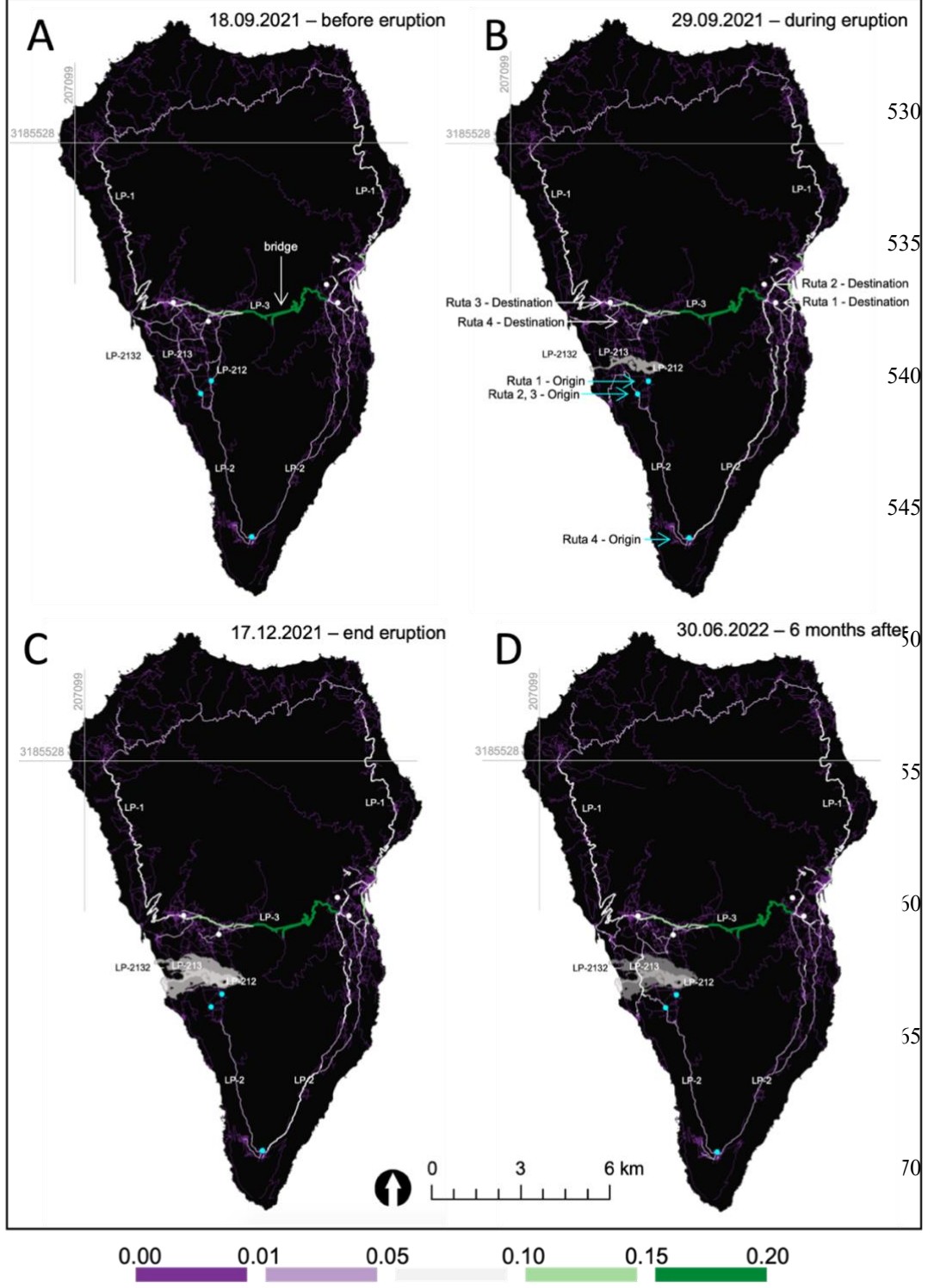

Figure 8. Betweenness centrality weighted by travel time (EBC-travel time) for lava+tephra scenario. A) before the eruption; B) during the eruption, C) end of eruption and D) 6 months after the eruption. Origin-Destination points for travel time estimations of 4 routes are shown in the panel A). Data source: Lava footprint taken from Copernicus Emergency Management Service (© 2021 European Union, EMSR546). Road network from © OpenStreetMap contributors 2021 (panels A-C), and 2022 (D). Distributed under the Open Data Commons Open Database License (ODbL) v1.0. Hillshade from ESRI. All maps using the Projected Universal Transverse Mercator (UTM) zone 28 on the World Geodetic System WGS84 datum (EPSG:32628).

### 4.2.3. Third order impact: systemic impact on the water system, agriculture and schools

The reduced functionality of the road network triggered important consequences to the island at all levels. Here we emphasize two main aspects to understand the magnitude of these cascading effects. First, there is a tendency to collocate CI components (e.g., power towers, electricity and telecommunications cables, gas and water pipes) along roads to ensure the accessibility (Rinaldi, 2001). All CI components directly in contact with lava were damaged, but also the accessibility to each system element (e.g., electric towers, water reservoirs, etc.) was interrupted due to the road cuts. Physical damage to north-south water irrigation channels provoked substantial decreases in farm productivity, which requires high volumes of water for operations. In particular, banana crops were highly impacted, as water was required to wash sedimented tephra from the crops. Figure 6C shows ~37 km of water pipelines (water channel LP-II, Punta Gorda - Las Manchas) and two reservoirs (Dos Pinos and Cuatro Caminos; 492,000 $m^3$ capacity) that were indirectly impacted by the first and second order impacts to the road network (Fig. 6C). An estimated area of ~25 $km^2$ of crops experienced systemic impacts, in addition to the 3.7 and 4.12 $km^2$ buried by lava and blanketed by tephra respectively (Fig. 6C in pink). Emergency measures to tackle the irrigation disruptions included stationing a water tanker, the reconnection of east-west water sources, and the installation of a desalination plant (shown with blue arrows in Fig. 6C).

Secondly, island-wide disturbances to daily activities were also experienced, not only due to evacuations and relocation of people and communities, but also to the travel time increases for work and school commuting. These disturbances exacerbated the direct impacts, and the consequences are still ongoing, 3 years after the end of the eruption. An illustrating example is the children migration process due to destruction and/or disruption of education centres. A total of 581 students and 87 professors were affected between September 2021 and January 2022, either by the direct impact or the reallocation-hosting process of children. Seven centres were directly affected with 295 children reallocated (3 schools totally destroyed, 4 closed due to the roads closure and/or evacuation restrictions), and 25 were indirectly affected because of hosting the evacuated children and/or on-line classes (Figs. 2 and 6C). From September 19 to October 18, educational activities took place remotely and schools closed in order to prepare a detailed Action Plan for Education Centres to ensure the security of students and staff. Until December 22, between 68 and 84% out of a total of 44 scholar days were on site, whilst between 16 and 32% were on-line for all municipalities concerned (i.e., El Paso, Los Llanos, Tazacorte, Tijarafe, Punta Gorda and Fuencaliente, Fig. 6C). On-line classes were associated with six days of confinements for bad air-quality, one for clean-up operations, and between 4 and 7 days for precaution in view of gas emissions caused by the lava entering to the sea (Fig. 2). In addition, travel time to arrive to the schools considerably increased with the subsequent psychological effects on children and parents.

### 4.3. Dynamic island connectivity: road network analysis

For this study, centrality network metrics, such as ECC, EBC-travel time and EBC-length, are used as proxies for island connectivity. A time-lapse animation of ECC over 58 dates (1 pre-eruption, 55 during and 2 post-eruption), considering the

scenario lava+tephra is displayed in the Supplementary Annexe 1. For the same scenario, Figure 8 shows the spatial distribution of EBC-travel time for four selected dates. The primary road LP-3 is a key bridging segment, joining the west and east of the island. Hence, this is the most central element of the road network, followed by segments in Los Llanos, the primary road LP-2 and the secondary LP-212 and LP-213 roads in the western flank of the island (Fig. 8A). On Sep. 29, when the lava flows reached the sea, EBC-travel time decreased for all roads proximal to the lava flow, whilst primary LP-2 and secondary LP-212 and LP-213 roads in the eastern flank of Cumbre Vieja became more central (Fig. 8B). By the end of the eruption (Dec. 17), low EBC-travel time values were re-distributed in Los Llanos, with no major changes in the rest of the island (Fig. 8C). The construction of a new road crossing the lava field, completed on May 31 2022 (Figs. 5D-E-F), induced a new centrality pattern where this new segment became highly central (Fig. 8D). In addition, the EBC-travel time for some secondary roads in Los Llanos and Las Manchas (north and south of the lava field) became also higher whilst for the eastern flank decreased again (Fig. 8D).

To explore the difference of the effects of lava and lava+tephra on the connectivity over time, the median values of both centrality indicators (calculated across the island network per date) are presented in Figure 9. A sharp decrease in connectivity is evidenced by the ECC trend on September 29 when the lava entered the sea and the western flank of the island was effectively segmented (Fig. 9A, red dotted line indicates the first lava flow into the sea). Other inflection points in ECC could be correlated with major changes on the width and length of lava flows (dashed lines in Fig. 9A; temporal variation of lava geometry in Supplementary Fig. S1). The EBC-length trends for both scenarios, lava and lava+tephra are very similar and show a significant decrease after the eruption started, decreasing until the LU sedimentation unit (Fig. 9B). During the MU unit, EBC-length for lava+tephra gradually decreased with respect to lava, demonstrating the effect of speed reduction on roads affected by tephra. By the middle of the UU unit, both scenarios strongly decreased with similar EBC values. Finally, the EBC-travel time was the most sensitive to the choice of the different speed reduction scenarios (RS, LS and HS, Table 2; Fig. 9C). HS and LS scenarios present the most continuous trends whilst the RS shows some inflections mostly in the middle of the LU unit. Importantly, during UU the EBC-travel time for the LS is higher than its value before the eruption. These two indicators clearly demonstrate the positive effect of the construction of the new roads (opened 2022 and 2023) on island connectivity, especially the EBC-weighted indicators.

In order to quantify the effect of connectivity loss on the associated systemic impacts, we calculated the change in travel time for four Origin-Destination (O-to-D) routes for selected essential service sectors (i.e., emergency, health, agriculture and education; Table 4; Fig. 8B and 10). Although we did not apply any topography or traffic impedance, travel time increases for all routes following a stair-step trend that is related to the lava first enters the sea (first dotted red line, Sep. 29, in Fig. 10), and tephra sedimentation units and sub-units (green LU, orange MU and purple UU lines in Fig. 10). O-to-D route travel times increased from 9-17 to 85 minutes by the end of the eruption, depending on the tephra speed reduction scenario applied (HS, LS, RS). Although the road network indicators are not highly sensitive to the speed reduction scenario applied (Fig. 9C), the

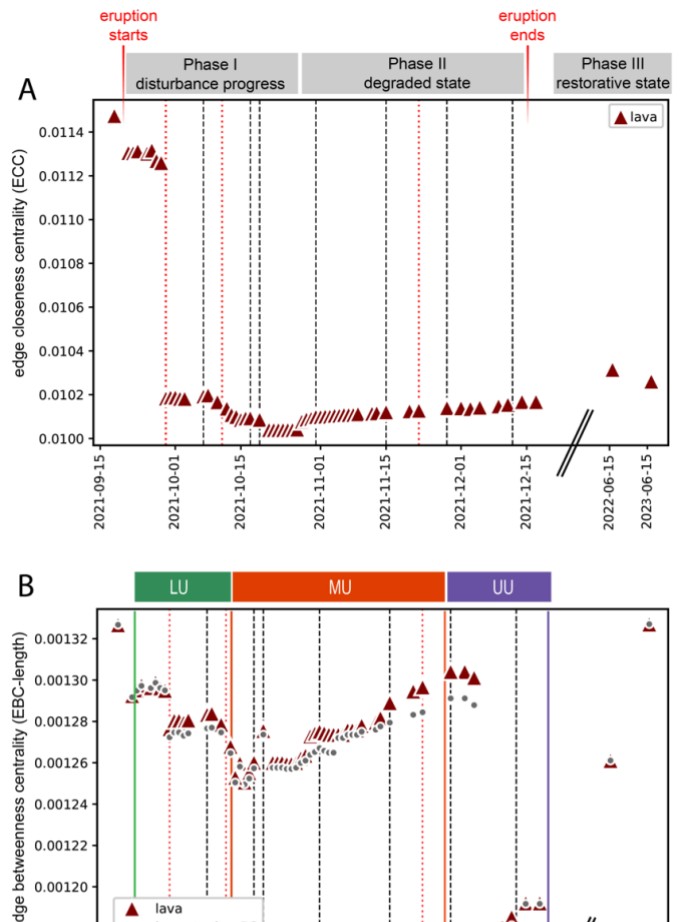

effect on travel time would strongly depend on these speed restrictions. This is particularly true for routes 2 and 3 (Figs. 10B, C, time-lapse animation in Supplementary Annexe 2). Total travel time across the four O-to-D routes is considerably reduced after the construction of the two new roads in 2022 and 2023, further demonstrating the restoration of island connectivity after this reconstruction phase (Fig. 5E, F).

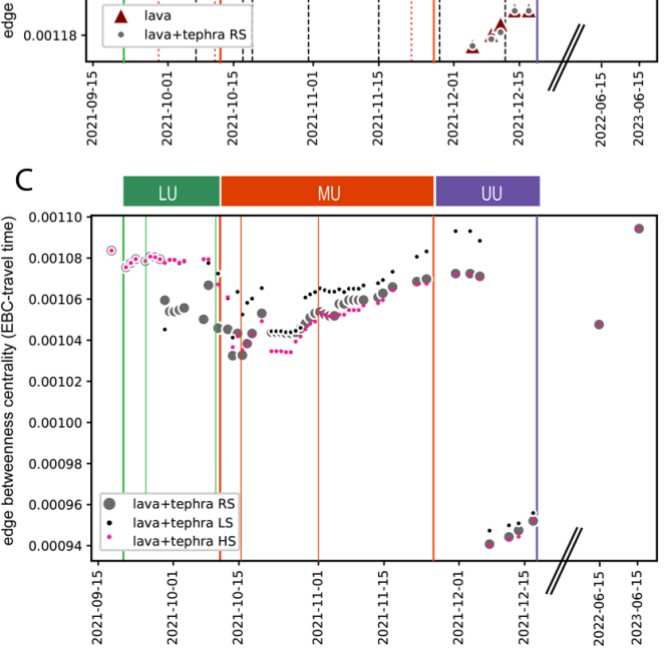

Figure 9. Road network indicators as a proxy of connectivity of the island over time. A) Median of the Edge Closeness Centrality (ECC) after removing edges and nodes affected by lava. Red dotted lines indicate lava flows entering into the sea and black dashed lines indicate major changes on the length and width of lava field geometry. B) Median of the Edge Betweenness Centrality weighted by length (EBC-length) to compare the difference on connectivity due to lava and to lava plus tephra by reducing the speed of edges affected by tephra (RS is Recommended Speed scenario of Table 2). C) Median of the Edge Betweenness Centrality weighted by travel time (EBC-travel time) to compare the effect of different speed restrictions scenarios (RS, LS and HS are Recommended, Low and High-Speed scenarios of Table 2, respectively). Green, orange and purple lines indicate the deposition of Lower (LU), Middle (MU) and Upper (UU) tephra units and sub-units.

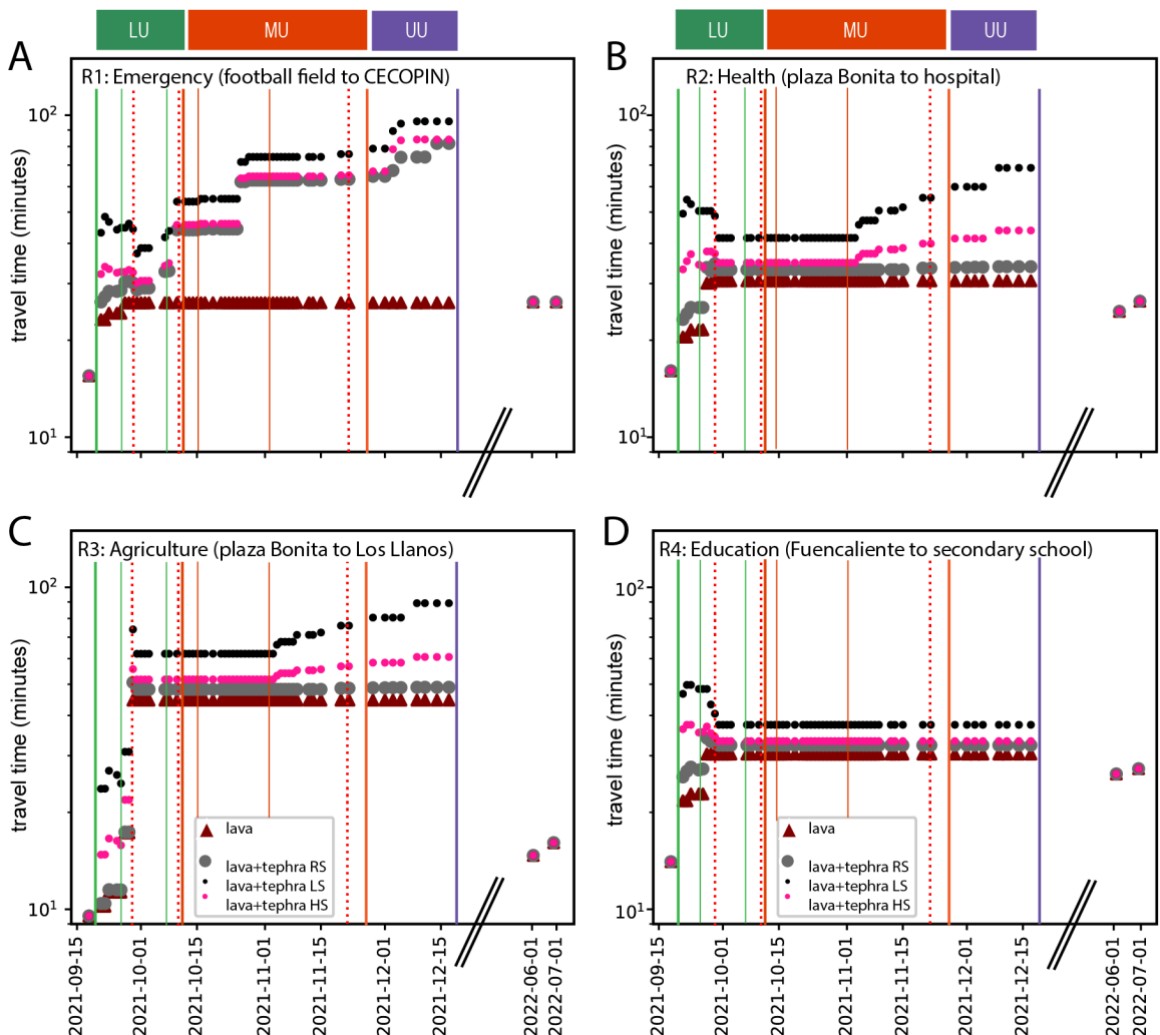

Figure 10. Travel time evolution for selected sectors for lava and tephra including 3 speed reduction scenarios (RS, LS and
HS). A) Route 1, emergency. B) Route 2, health. C) Route 3, agriculture, and D) Route 4, education. Red dotted lines
indicate lava flows entering into the sea and green, orange and purple lines indicate the deposition of Lower (LU), Middle
(MU) and Upper (UU) tephra units and sub-units. See Table 4 and Figure 7A for location of these Origin-Destination points.

## 5. Discussion

Volcanic eruptions generate significant cascading impacts that affect the functionality of societies, particularly challenging

decision-making, response and recovery capacity. The integrative PEIA of the 2021 Tajogaite eruption assessing the causal

order of impacts due to compounded volcanic hazards (i.e., lava and tephra fallout), complemented with a road network

analysis, provides a robust framework to quantify critical infrastructure performance in case of volcanic eruptions. This

approach helps understand how systems are connected and ultimately informs the selection of effective and rapid mitigation and recovery actions for future risk reduction.

## 5.1. Relevance of an integrative PEIA based on causal order of impacts for risk reduction

PEIAs are essential for identifying the drivers of volcanic impacts on the current capacity of interconnected systems and thus, for informing decision-makers in the case of future events. Whilst traditional PEIAs offer a broad qualitative assessment, a forensic approach allows for the identification of root causes, enhancing the forecasting of potential impacts, and improving the policies in land-use and CI performance (Ferreira et al., 2023). The integrative forensic framework, introduced by Dominguez et al. (2021) and applied here, investigates complex cascading chains of volcanic impacts by discretising 3 causal orders, physical, functional, and systemic. This framework was proposed for assessing the impacts of the 2011 Cordón Caulle tephra fallout and ash remobilisation on electricity systems (Dominguez et al. 2021). Similar approaches with a reliability engineering component have been developed for other natural and anthropogenic hazards, e.g., earthquakes (Pitilakis et al., 2013); nuclear plant disruptions (IAEA, 2015); groundwater pollution (Garcia-Aristizabal et al., 2019); and incident analysis for gas resources (Garcia-Aristizabal et al., 2017).

In the case of the 2021 Tajogaite PEIA, compound volcanic hazardous phenomena and products (Fig. 1) affecting the socio-economic configuration of La Palma (Fig. 3) have been carefully analysed to schematize an overall view of impacts/response (Fig. 2) and identify major cascading effects, that were associated by all the stakeholders with the road network disruption (Fig. 6, Table 5). The same logical thinking can be applied to any other CI system. Characterised by an insular closed network with limited space, a severe disturbance of the ground transportation occurred due to the location (inland ~7 km from the eastern coast), the long duration (3 months), and the magnitude (>12 km$^2$ of lava field, >190 km$^2$ of tephra blanket and 187 m high tephra cone) of this eruption. As a result, all activities across the island were disturbed. The destruction of the primary LP-2 road (connecting the commercial centre of Los Llanos and the south, Fig. 6) during the first few hours of the eruption rapidly reduced the connectivity of the island. The impact sequence initiated by the physical damage of roads due to lava flows (1st order; Fig. 6A) was followed by the loss of functionality of the road network due to the combination of lava flows, tephra fallout and restrictions associated with the exclusion zone (2nd order; Fig. 6B), which then triggered systemic impacts to socio-economic activities dependent on ground transportation (such as water, agriculture and education sectors) (3rd order; Fig. 6C).

A logical thinking involving a deductive (backward) reasoning, identifying the root causes of impacts, and an inductive (forward) reasoning, identifying the subsequent cascading consequences was essential to define five road functionality scenarios (FL-0 to FL-V, Fig. 7A) and reconstruct the associated temporal and space scales (reflected in Table 5). However, with the limited data available, the only chronological reconstruction is the road closure due to lava (FL-IV) thanks to the real-time lava runout monitoring conducted by PEVOLCA and supported by the rapid mapping of Copernicus. For this reason, the

evolution of functionality scenarios presented in Fig. 7B is rather constant over time, with the exception of the lava scenario (FL-IV), whose spatiotemporal evolution can be reconstructed. Concerning the reduced service due to tephra accumulation (FL-II), we found that even a few millimetres of tephra were enough to affect nearly the entire area eventually covered by the full deposit (>190 km$^2$) since the beginning of the eruption. As a result, the road length impacted by FL-II remained almost constant throughout the event, regardless of the total accumulated tephra (Fig. 7B). This indicates that a minimal tephra load was sufficient to trigger FL-II conditions. However, a detailed analysis of tephra thickness evolution over time is needed to assess varying levels of disruption. In Figure 7C we used the deposit reconstruction model presented by Bonadonna et al., (2022) and Biass et al., (2024) to apply the existent IS for tephra disruption associated with tephra thickness (Blake et al., 2017a). At the beginning of the eruption, roads were primarily determined to be IS1a (minor skid resistance, reduction possible, 0.01-0.1 cm and 0.5-10 cm of tephra) and IS1b (skid resistance, reduction likely, 0.1-0.5 cm tephra). Interestingly, Blake et al., (2017a) found that skid resistance increases in the range of 0.5-10 cm of deposited tephra, meaning IS1a is likely to be constant once this thickness accumulation is reached (Fig. 7C). Contrastingly, IS1b follows a peculiar pattern, steadily decreasing to zero km of roads affected by the end of the LU unit, and increasing again from the beginning of the MU unit until first days of November (Fig. 7C). This pattern could be explained by the fact that wind direction changed towards the south during the sedimentation of the sub-unit MU1, whilst most of the LU was deposited in a NE-SW dispersal axis (see Fig.7 in Bonadonna et al., 2022). This change in wind direction implies the accumulation of few tephra (0.1-0.5 cm) in the south explaining the presence of IS1b until the end of MU2-5 sedimentation, which coincide with November 1st. Trends of IS2 and IS3 follow a rapid monotonic increasing during the sedimentation of the LU unit and are more constant during the MU and UU units, being these two units less intense, in agreement with lower mass eruptions rates calculated by Bonadonna et al., (2022). It is important to highlight here that these IS are overestimated since no clean-up operations have been considered. In fact, clean-up operations and a rapid construction of a road bypass in the south of the lava field, during the three months of the eruption were implemented by PEVOLCA to ensure emergency activities, but, given that details on these measures were not available, our analysis is based on the no-cleaning assumption.

Although lava flows generally cause severe road disconnection due to its potential of total destruction, tephra is highly disruptive and often provokes driving difficulties, traffic jams (Blake et al., 2017a; Wilson et al., 2017) and fatal accidents (Blake, 2016; Wardman et al., 2012; Hayes et al., 2022). In La Palma no traffic accidents have been directly associated with tephra fallout; nevertheless, significant traffic jams, congestion and complicated circulation were reported within the exclusion zone and down to Fuencaliente (Fig. 6B). However, it is difficult to distinguish the source of disruptions in this road segment, since public works are very frequent, even before the eruption. Although developed IS for tephra only involve skid resistance and marking cover as a function of tephra thickness, the interaction of tephra and roads involve a complex relation of various physical parameters such as the effect of grainsize, abrasion, ash settling rate, soluble components, wetness, wind conditions, road type, asphalt and painting characteristics, among others (Blake et al., 2017a). Indeed, the important contribution of loss of visibility for driving, associated with visual range or ash settling velocities cannot be easily measured or estimated in real

time in the field, and its relationship with tephra thickness is still under experimental research (Blake et al., 2018). However, all these aspects generally trigger an instinctive speed reduction, in a similar way as with fog (e.g., Brooks et al., 2011). Currently, there is no standard guidelines to correlate tephra IS and speed reduction measures. The safest approach is to apply the recommended regulations of each country during bad weather conditions that could be adapted for the particular

characteristics of tephra sedimentation and ash remobilisation. In this study we attempt to correlate theoretical IS with the levels of functionality defined by the DGT of Spain, by deducting scenarios from technical reports of road closure or traffic disruption under adverse weather conditions (Table 2). However, a more accurate analysis of disruption records would shed light on how decisions are made and how recommended speeds or driving behaviours are established during a volcanic crisis.

**5.2 Importance of road network analysis for PEIA: potential and limitations**

The integrative PEIA based on causal order previously discussed is useful to delineate cascading paths and discretise each contributor of impact or response in a versatile way. However, this approach cannot quantify the loss of connectivity of the road network and the consequences on other systems across space and time. The advantage of using graph theory to describe the road connections based on the edges/nodes relations, and how these connections change over time, has an enormous

potential to analyse CI performance. The detailed network analysis provides a robust and objective measure of indicators for the whole island that are useful to i) identify the central elements of the network, and hence prioritise mitigation measures for future eruptions (Fig. 8); ii) quantify the spatiotemporal evolution of indicators as a proxy of loss of connectivity throughout the eruption (Fig. 9); and iii) estimate the systemic impact on other sectors in terms of travel time (Fig. 10). Although all selected indicators clearly evidence the disruption on the network due to the eruption, each of those provided different

information and insights. For instance, closeness centrality (ECC, Fig. 9A) evidences an abrupt change at the beginning of the eruption with a more constant evolution subsequently, that could be interpreted as different system performance phases named as disturbance progress, degraded and restorative states (Fig. 9A). Although never explored before in volcanology, these phases are being widely used on the analysis of electricity and gas functionality in case of extreme weather or cyber-attack events (Kehkashan et al., 2022; Xie et al., 2022; Kandaperumal and Srivastava, 2020; Hossain et al., 2021). In terms of resilience

metrics, identifying these phases is fundamental to quantify levels of performance and how the systems respond to and adapt from volcanic events. On the other hand, the edge betweenness (EBC) allows a comparison between various impact scenarios, in our case, lava and lava+tephra since calculations are based on a different edge weight (e.g., length, travel time, Table 3), and therefore edge speed restrictions due to tephra can be considered. EBC-length evolution for La Palma follows a stair-step decreasing trend rather than monotonic that could be easily correlated with major changes of the hazards' geometry; especially

the width and length of the lava field (Fig. 9B and supplementary S1). In fact, EBC-length results corroborate the idea that major contribution on the connectivity loss of the road network is dictated by the lava (main decay pattern) and secondarily by tephra, which follows in turn the main lava trend.

Finally, the EBC, weighted by travel time, is particularly interesting when analysing various speed restrictions scenarios (low-LS, high-HS, recommended-RS, Fig. 9C, Table 2). Our results suggest that the LS scenario, although conservative, forced EBC-travel time to be higher than before the eruption which is difficult to interpret. RS and HS scenarios are contrastingly more consistent. In any case, the effect of using either of the speed scenarios does not result in significant variations of the EBC median, in comparison with the main pattern dictated by the lava impact. Despite this EBC-travel time trend not being very sensitive to the selection of speed scenarios, its effect on the actual travel time from one point to another (O-to-D points, Table 4) is significative (Fig. 10). LS scenario increases up to 60% the travel times of reaching a point in comparison with RS scenario, especially for the routes 2 and 3 given the large difference in the path prior and after eruption (Fig.10, Supplementary Annexe 2); and HS scenario results in mid values between the RS and LS scenarios. Ideally, a calibration should be carried out based on the actual time necessary to drive from O-to-D points to select the speed restriction scenario that better adapts to the conditions in La Palma. To do this, an impedance factor accounting for topography, traffic jams, weather conditions, vehicle type would be necessary. Regardless of the speed scenario selection, the estimation of travel time for selected O-to-D points represents an objective measure to quantify the systemic consequences (3$^{rd}$ order) in other sectors as a result of increasing road travel time. Here we have explored the routes of specific interconnected systems in La Palma (i.e., emergency, health, agriculture and education, Table 4), but any other O-to-D can be analysed.

Quantifying cascading impacts of interconnected and interdependent CI exposed to volcanic eruptions require sophisticated tools. The approach developed in this study combining forensic techniques with network analysis indicators provides a robust framework that is pioneering in volcanology. Network analysis has been used to determine the resilience of CI networks facing various hazards based on the comparison of centrality indicators as a response to disturbances through node/edge removal (e.g., railways, Bhatia et al., (2015); transportation, Rouhana and Jawad (2025); telecommunications Alenazi, (2023)). However, the temporal evolution of centrality to describe performance before and after crisis has been mostly applied in the financial sector to identify systemic interconnections of institutions (e.g., Kuzubaş et al., (2014); Rossi et al., (2018)). The versatility of this approach allows the applicability to any region facing a disturbance since graph indicators are calculated with the principle of node/edge removal regardless the hazard footprint. However, CI of modern societies are complex networks that need graph metrics whose mathematical expressions require specific structures (i.e., edges, nodes, relationships). In the case of road networks, these structures are relatively easy to access thanks to the OSMnx package that uses the OSM database (Boeing, 2017). However, the same approach could be applied to any other CI network once the graph is built (Barthélemy, 2011), resulting in a promising set of tools for future volcanic impact assessments. Future research should focus on the selection of indicators with a physical meaning, which must be developed as we explore better this approach in post- or pre-event impact assessments. The proposed methodology could be therefore complemented with modelling of future scenarios to inform land use planners and stakeholders where are the most critical sectors of CI networks. It is important to highlight that the temporal evolution of the road network presented in this study was possible thanks to the high-resolution (in space and time) of lava footprints, available from Copernicus and activated by PEVOLCA. The daily monitoring of lava evolution was

crucial to understand variations of road network indicators associated with specific dates (e.g., lava reaching the sea, geometric changes of the lava field) that could completely shift the final impact scenario. We therefore recommend a monitoring from ground and space of the different volcanic products emitted during a volcanic crisis, as frequently as possible, with the aim of providing sufficient and accurate data to improve the impact assessments of future volcanic eruptions.

## 5.3. Recovery and reconstruction of the road network

Internal reduction of island connectivity due to the destruction of the primary (LP-2) and secondary (LP-213) roads, required ingenious and rapid reconstruction strategies. The eruption ended on December 13, and the construction of a new road started on December 25 aiming to cross the lava field (Fig. 5D), which had averaged maximum thicknesses between 12 and 70 m (Cabrera García, 2023). The rapidity of this reconstruction measure in relation to a road network is pioneering worldwide. Indeed, the severe impact on all daily activities of the island rose a social discomfort that required rapid measures (Carracedo et al., 2022b; Brusini Dominguez, 2022; Troll et al., 2023; Rodríguez-Pérez et al., 2024). The reconstruction of a new road to reconnect the north and south of the western side of La Palma (Fig. 5D) involved important challenges, not only in terms of physical aspects (e.g., extremely high temperatures, lava thickness and cavities, type of pavement materials), but also in terms of political (e.g., hierarchical governance levels for each type of road) and cadastral aspects. Indeed, according to Spain law, even when buildings are destroyed by a natural event, the land continues to belong to its owner, and this increased the complexity of designing a new route segment (Cabrera García, 2023). The new LP-213 road of La Palma, built with eco-materials (e.g., volcanic ash, lime, salt water), was operational on May 31, 2022 (with timing and vehicle type restrictions), 6 months after the end of the eruption (Figs. 5D-E-F). A second road, the LP-2132 was also opened by June 2023 (Fig. 8D). Interestingly, the effect of this reconstruction measure is also evidenced by all the road network indicators, clearly showing how the new segment of the LP-213 became central (Fig. 8D) and how the connectivity increased during the restorative phase (Fig. 9).

## 5.4. Insular and monogenetic settings

Volcanic eruptions in insular environments can aggravate the loss of connectivity of an entire community. Whilst it is true that, due to the multi-hazard aspect, volcanic eruptions are generally highly disruptive in comparison with other natural hazards (Bonadonna et al., 2018), it is also true that insular contexts, and hence, geographically closed systems with low redundant and circular networks, introduce even more challenges for crisis management and risk reduction (Castanho et al., 2019, 2020). This is an important aspect considering that most offshore islands in the world are volcanic. For example, the East Pacific Island Arc including 45% of the world volcanoes (Kirianov, 2007) is also one of the most populated areas of the world (Cottrell, 2015), with an averaged population density of 700 people per $km^2$ within 20 km around volcanoes (Freire et al., 2019). Generalized island disconnections depend on the size of the island and urban patterns (redundancy and fabric of road networks). Small islands are particularly vulnerable due to their remoteness, spatially limited (and often fragile) economies and resources

(Turvey, 2007), and scarce space to reconvert land-use (Castanho et al., 2019; Gómez et al., 2020). Road network disconnection is of the utmost importance for crisis management where evacuation processes are imminent. For instance, Jenkins et al. (2017) described the high dependency of all communities to a single road in the Fogo island during the 2014-2015 eruption, and highlighted that the best risk reduction against lava flows remains the relocation of infrastructure. However, this relocation is practically impossible if we add the complexity of monogenetic volcanism where urban areas are scattered around the whole volcanic field, as it is the case in La Palma (Marrero et al., 2019). In this context, volcanic impacts are difficult to forecast, in large part due to the highly uncertain eruption source location, and difficult to mitigate, due to the limited physical space and fixed infrastructure configuration of small islands. To illustrate this effect, Fig. 11 displays a comparison of three road network indicators (ECC, EBC-length and EBC-travel time) by using the current road network and historical lava footprints, and consequently different vent locations and lava field surfaces (triangle size in Fig. 10, Supplementary Table S1). These three indicators show that San Juan/Hoyo Negro would be the most impactful scenario, together with Tajogaite, if a similar eruption occurred today. Whilst the road disconnection of Tajogaite is expected, being the largest lava field of historical eruptions of La Palma ($>12$ km$^2$), San Juan/Hoyo Negro, although small (4.5 km$^2$) shows lower ECC and EBC values because of its runout in 2 directions (western and eastern flanks of Cumbre Vieja, Fig. 1A) that would provoke a generalized disruption of the south of the island. Contrastingly, Teneguía and San Antonio scenarios, being the southernmost eruptions, would have the less impact on connectivity. Interestingly, Tacande scenario, located in the north towards Los Llanos, shows a low ECC value, since it would cut the highly central LP-3 (Fig. 8A); but high EBC values, probably because there are several alternatives (multiple shortest paths) parallel to the LP-3 in Los Llanos. This scenario comparison reveals that vent location and the lava runout direction and extent play a crucial role on the final impact on roads, which ultimately would dictate the subsequent cascading impacts on socio-economic sectors. This network analysis throughout the different historical eruptions of La Palma clearly demonstrates the enormous potential of graph theory and indicator analysis in quantifying connectivity and dependency of CI. However, it is important to note that this methodology applies primarily to closed systems (i.e., islands) where the median or mean of graph indicators can be directly calculated. In the case of open systems, such as continental regions, averaging indicators would require the definition of boundaries either by country or by specific case study areas.

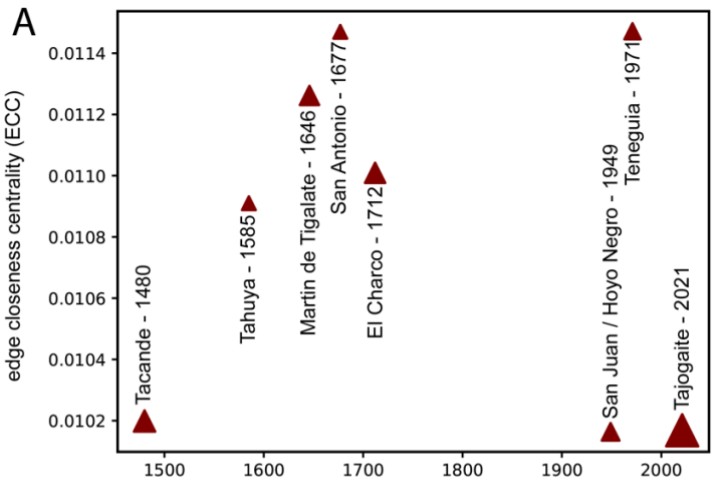

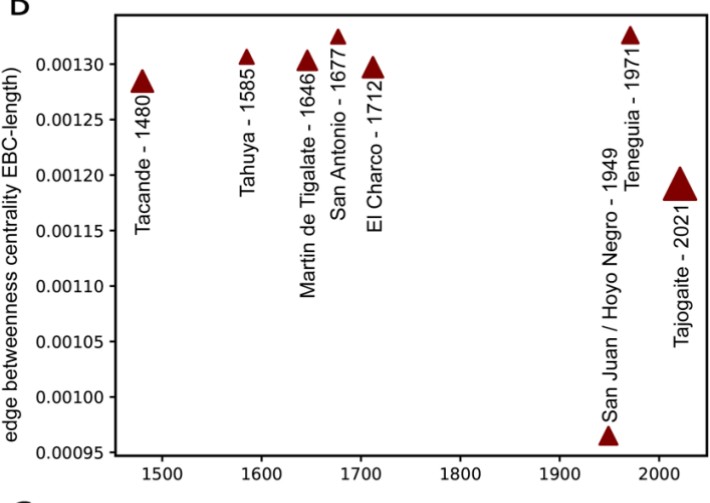

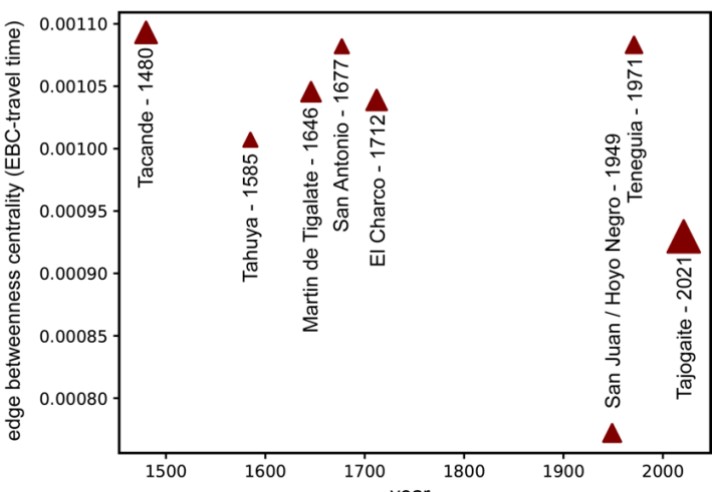

Figure 11. Comparison with historical eruptions lava fields. Road network indicators calculated by removing edges and nodes of historical lava fields assuming the current infrastructure in La Palma. A) Median of the Edge Closeness Centrality (ECC). B) Median of the Edge Betweenness Centrality (EBC) weighted by length (EBC-length). C) Median of the Edge Betweenness Centrality (EBC) weighted by travel time (EBC-travel time). Triangle size varies as a function of lava field surface (see Fig. 1A for location).

## 6. Conclusions

The functionality of modern and interconnected societies exposed to volcanic eruptions strongly depends on an equilibrium of governance, socio-economic sectors and resilient infrastructures with well-developed volcanic risk management plans. Decision making is especially important in case of imminent eruptions, as well as the degree and efficiency of response and subsequent recovery measures. The 2021 Tajogaite eruption demonstrated that small-moderate size eruptions can provoke important cascading effects when long-lasting compound hazards are combined with low-redundancy and circular infrastructure, typical of insular environments. These cascading impacts largely compromise the vital societal functions that rely on the functioning of critical infrastructures. Impact assessments therefore require the development of comprehensive approaches capable of including all relevant components of systemic risk.

The PEIA developed in this study, combining forensic techniques with network analysis tools, provides a solid methodology to assess various orders of cascading impacts, often associated with volcanic eruptions. On the

one hand, the forensic approach, based on causal order of impacts (in-depth root cause and consequence analysis) allows the identification of main drivers of cascading paths that can be described by logical inductive and deductive tools (e.g., event trees). The versatility of this methodology to answer key questions such as the "what, where, when, and how" impacts (and response) occur has an enormous potential when quantifying volcanic multi-hazard impacts in a causative perspective, and hence the lessons learnt from past eruptions become more evident. On the other hand, network analysis provides an objective and robust measure of connectivity of critical infrastructures which in turn influence the interdependencies with other systems. Network analysis has been widely used on reliability of critical infrastructures but is poorly explored in volcanology. The utility of the OSMnx package (based on NetworkX) to analyse the volcanic impact on road network opens a new research field in volcanic risk assessment.

Despite volcanic disruption metrics being still subject of discussion, the presented PEIA proposes the characterisation of three orders of impact: physical damage, loss of functionality and systemic impact. The loss of functionality has been described by 5 levels: no loss of functionality, disrupted service, reduced service due to tephra accumulation and due to exclusion zone restrictions, and permanent closure. Reduced service due to tephra impact has been subsequently sub-divided in 5 impact states as a function of tephra thickness and were correlated to the specific traffic/circulation functionality levels of Spain (i.e., no restriction, conditioned, irregular, difficult, interrupted circulation). The causal order of impacts, core of the presented PEIA, could be applied to any other volcanic environment, by adapting the specific scenarios of functionality and systemic impact to each specific case and/or country.

More specifically for La Palma, the 2021 Tajogaite eruption highlighted that monogenetic volcanism involving compound hazards of long duration is highly critical in insular environments. The geographic and geologic contexts of Cumbre Vieja imply that volcanic impacts are difficult to forecast, mitigate and avoid due to large uncertainties on source location and limited space to relocate infrastructures or reconvert land uses. Our results demonstrated that the location of future eruptions would dictate a range of expected impact scenarios. Exploiting the potential of network analysis with modelling of vent-opening locations would provide key information for emergency and risk management. Consequently, risk reduction strategies should focus on implementing sustainable risk management plans tailored to the needs of each political unit of the territory (i.e., municipality, island, province, community, state), as is the case of Spain. Specifically, the volcanic emergency plan, PEVOLCA (updated on 2018), was applied for the Tajogaite eruption, demonstrating that the main objective of protecting human life and the coordination of all the different stakeholders was achieved, although important lessons were learnt that allow further improvements (Brusini Dominguez, 2022). Even though discussing these improvements are beyond the scope of this paper, we would like to emphasize the relevance of PEVOLCA action of requesting the rapid mapping of the Copernicus Emergency Management Service, which was fundamental for this PEIA analysis. Indeed, the high temporal and spatial resolution of lava footprints and interpolation of tephra blanket deposit drove to a detailed chronological PEIA allowing the

correlation of specific hazard characteristics (e.g., lava path towards the sea, changes of tephra eruption rate) with inflection events of the road disruption.

**Acknowledgements**

The authors are deeply thankful to the communities of Los Llanos, El Paso, Tazacorte and Fuencaliente for sharing their experience during the devastating volcanic crisis and reconstruction period after the 2021 Tajogaite eruption. Special thanks to the PEVOLCA committee for permission to restricted areas to conduct fieldwork during the eruption, and all local stakeholders for assistance and support of this research. We are grateful to the Instituto Volcanológico de Canarias (INVOLCAN) for the logistics and support in the field. We are especially grateful to Montserrat Roman and Amilcar Cabrera for insightful discussions on crisis and risk management, and to Jonathan Lemus and Sebastian Possos for network analysis.

**Funding**

This research has been supported by the Swiss National Science Foundation Grant #200020_188757. Ethical approval for engagement with volcanic risk management stakeholders was granted by the University of Geneva for the funding project associated with this study.

**Author contribution**

LD, CF, SB and CB designed the project and the methodology under the funding acquired by CB. LD, CF, MPRH, LSDI, CB and NP have performed field work. LD and SB performed network analysis. LD, CF, SB, AW contributed to data analysis. LD drafted the manuscript. All authors contributed in discussion and final compilation of the manuscript.

**Data availability**

The data that support the findings of this study are openly available in the Zenodo repository at
https://doi.org/10.5281/zenodo.17543235

**Competing interests**

The contact author had declared that none of the authors has any competing interests.

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
