# Peer review of "Quantifying cascading impacts through road network analysis in an insular volcanic setting: the 2021 Tajogaite eruption of La Palma Island (Spain)"

_EGUsphere, 2025_

## Referee Comment (RC1)

**REVIEW**

**Title: Quantifying cascading impacts through road network analysis in an insular volcanic setting: the 2021 Tajogaite eruption of La Palma Island (Spain)**

**Decision: Major revisions**

**General review:**

This paper presents a detailed post-event impact assessment (PEIA) that highlights the compound nature of volcanic hazards and their far-reaching consequences on interconnected infrastructure systems. By adopting a forensic approach and applying graph theory-based network analysis, the study quantifies both direct physical damages and indirect systemic disruptions—particularly within transportation, emergency services, agriculture, and education sectors.

The objective of the work is clearly defined, and the case study is of utmost importance for the scientific community. The study is grounded in the specific context of the 2021 Tajogaite eruption, and this focused approach allows for a nuanced and thorough examination of local impacts and system responses. The depth of analysis provides clear value for understanding complex interactions in disaster scenarios. Nonetheless, the strong emphasis on a single case may somewhat narrow the study's relevance to broader contexts. A more explicit connection to similar challenges in other regions would further enhance the paper's applicability and reach. The case-study of La Palma is undeniably meaningful but I would suggest giving a bit more emphasis to broader implications on the aspects that would make it usable in another region and context.

I think you approach could gain relevance if discussed against similar approaches developed for other natural hazards and for tacking the deriving emergencies, as well as for multi-hazard scenarios. A wider comparative perspective or a discussion linking the results to other active volcanic areas or areas prone to other hazards (e.g. landslides) would enhance the study's impact and utility for a more diverse audience. Expanding the contextualization would help readers better understand how the presented methods and insights could be adapted to different geographic or socioeconomic settings.

With regards to the methodology, I suggest to discuss a bit more how these ex-post assessments can help future risk management. How can these be included in capacity planning, emergency management and in the definition of future exposure and impact scenarios? I really like your idea of using past lava flow locations and this might be discussed explaining how this could be complemented (e.g. with modeling) to provide future scenarios to land use planners and infrastructure managers. You already started discussing this in line line 907-915 but I think a bit more emphasis could be given to this.

Finally, I think the manuscript could benefit from an overall polishing to avoid repetitions which are sometimes confusing for the reader and perhaps distract from the main message (see specific comments below).

Overall, I congratulate the authors for their rigorous and insightful work, which makes a valuable contribution to disaster impact analysis; with the suggested enhancements to broaden its contextual reach, this study has the potential to become a significant reference for the scientific and risk management communities.

**Specific comments:**

**Introduction**:

In this work it is somehow unclear why you use the term systemic or cascading impacts, and if you are tackling both. In particular, you mention in line 93-95 that the cascading effects are triggered due to the interaction between roads and other exposed assets, so, to my understanding, we are talking about the induced impacts due to road network disruptions. However, other systemic impacts are also described. I think the reader would benefit for a clarification in the introduction so it's clear what is meant.

Lines 100-104: It could be worth mentioning the study from Scaini et al. 2014 on the loss of functionality in terms of physical and systemic vulnerability induced by compound volcanic hazards in Tenerife on exposed assets including roads.

Fig.2: The primary road (black line) should be included in the legend, in addition to the caption

**Methodology**:

Line 293-294: I think the reader would benefit from a broader description of the involved stakeholders and how they were involved, and for example if the same stakeholders were participating to all the encounters. Also, do the mentioned discussions only include structured interviews or is there also a different kind of interaction (e.g. group meetings)? Did they show you the location of the impacts using a map, or how did they identify the impacted areas and/or the location of the assets? Also, In line with my comment on the generalization of your approach, how do you suggest to adapt the interview questions to different contexts? How was this done passing from Dominguez et al., 2011 to this context?

Line 300: I think this sentence explains very well what is done and could be moved to the initial methodological part rather than the data acquisition section.

Table 1: why 'disrupted' is FL-I and 'reduced' is FL-II? How were these functionality levels defined? Were they all observed in the case-study?

Line 343-346: The information in the road network data should be moved to the data section

**Results:**

Section 4.1 mostly presents the impacts in a narrative form and it's hard for the reader to distinguish if these information were collected from the presented methods, or if they are part of the context and the already available information. In the second case, these narratives could be provided at the beginning to explain to the reader what is the situation, while the results should present your original findings.

Lines 385 -389 and Supplementary material on impacts to other critical infrastructure systems: I am unsure why these impacts are summarized here, if it's to demonstrate the impacts due to the volcanic hazards or to the road closure. This li ks back to the question on the definition of cascading and systemic impacts, which I think could be made clearer in the text.

Table 5: I think this table is very relevant to understand what are the results of this work.
Row 2, column 5: 'Highly disruptive' is a bit generic
Row 7, column 4: is the surface missing?

Section 4.2.2: The initial part is a repetition of the method which should be already explained in the methodology section.

Section 4.2.3: Similarly to section 4.1 there is a broad description of what happened but it is somehow unclear what fraction of this information was gathered using your methods (e.g. interviews) and which information was already available in advance. Even in this second case, if you reinterpreted this based on the impact chains that you identified and on your methods, this can be part of the results, but it should be made clear to the reader.

**Discussion:**

Line 730-740: I would include in the introduction some background on the ex-post analyses and the forensic approach in the field of disaster risk reduction, with literature references (e.g. AM Ferreira et al., 2023; F. Atun, 2024), in order to highlight the relevance of you work beyond the case-study. Also, this could be discussed to show what are the improvements of the approach that you propose.

Line 745-746: 'identify the most impactful cascading effects': Is this shown in the figures? Or was this part of the replies that they gave you to the interviews?

Line 760-764: It would be relevant to explain what kind of data would support a deeper and more extended deductive/inductive reasoning. This could help other researchers interested in applying this tool in other areas where such data might be available.

Line 765-768: This sentence is not super clear to me, could you rephrase? Does it mean that, since the majority of the tephra deposition happened in the first phase and that the impacted road length are almost constant during the following phases?

Given that you comment on different CI (e.g. water networks) and given the presence of the water reservoirs that are, both here and in other Canary Islands, so relevant for local activities, I would suggest adding some references in the introduction about the exposure and vulnerability of these (e.g. Stewart et al., 2006, 2009).

Fig. 6C: Is it systemic or cascading?

Line 814-815: how much of this analysis can be done a priori? e.g. analyzing the existing network and its potential pitfalls using a combination of scenario modeling and network analysis?

Apart from the road type (and associated speed), did you account for the road path (e.g. if it is mostly straight or curved) and the pavement type? Also, how different could this scenario in case of different weather conditions (e.g. under heavy rainfalls), especially in areas served by unpaved secondary or tertiary roads?

Do you foresee to tackle the lack of traffic data in some other way (e.g. using phone data, such as in the work of Yabe et al., 2022; Wilson et al., 2016; Giardini et al., 2023)? Or, if not, shall this be addressed in the future?

**Conclusions:**

Following my comment on the results, I think the conclusions are a bit generic and could be made stronger by giving more emphasis to the specific results achieved with the analysis (e.g. from table

5, fig. 10-11). In particular, from Fig. 10 it seems that the travel time increases a lot for R1, and this might set a priority for emergency managers.

**Minor comments:**

If possible, I would suggest to translate the tables in S2 into english

I'm not a big fan of acronyms and sometimes I feel their use could be reduced in favor of readability.

**Supplement:**

In the section regarding air traffic disruptions, I think the Sara et al. Reference Barsotti et al.:
Sara, B., Simona, S., Giovanni, M., Alicia, F., Aline, P., Georgios, V., de Zeeuw van Dalfsen, E., Lars, O., Adriano, P., Jean-Christophe, K., Susan, L., Rita, C., Mauro, C., Jordane, C., Charlotte, V. B., Mauro, D. V., de Chabalier, J. B., Teresa, F., Fontaine Fabrice, R., Arnaud, L., Rui, M., Joana, M., Roberto, M., Anne, P. M., Jean-Marie, S., Ivan, V., Kristín, V., Samantha, E., and Giuseppe, S.: The European Volcano Observatories and their use of the aviation colour code system, Springer Berlin Heidelberg, https://doi.org/10.1007/s00445-024-01712-0, 2024

Also, on the tephra-induced impacts there are more references that could be provided.

With regards to the selection and classification of exposure assets (S3), I think it should be discussed and compared with other existing classification and taxonomies (e.g. the GED4ALL) and their adoption for exposure assessment in other study areas. Exposure is inherently multi-hazard, so the approach would benefit from the adoption of a more general taxonomy. Also, I suggest discussing how local-scale methods such as the one proposed here can be integrated in the overall multi-hazard risk assessment.

---

## Author Response (AR1)

**Dominguez et al. in review: Response to reviewers**

**Reviewer #1**

**Title: Quantifying cascading impacts through road network analysis in an insular volcanic setting: the 2021 Tajogaite eruption of La Palma Island (Spain)**

**Decision: Major revisions**

**General review:**

This paper presents a detailed post-event impact assessment (PEIA) that highlights the compound nature of volcanic hazards and their far-reaching consequences on interconnected infrastructure systems. By adopting a forensic approach and applying graph theory-based network analysis, the study quantifies both direct physical damages and indirect systemic disruptions—particularly within transportation, emergency services, agriculture, and education sectors.

**No response needed.**

The objective of the work is clearly defined, and the case study is of utmost importance for the scientific community. The study is grounded in the specific context of the 2021 Tajogaite eruption, and this focused approach allows for a nuanced and thorough examination of local impacts and system responses. The depth of analysis provides clear value for understanding complex interactions in disaster scenarios. Nonetheless, the strong emphasis on a single case may somewhat narrow the study's relevance to broader contexts. A more explicit connection to similar challenges in other regions would further enhance the paper's applicability and reach. The case-study of La Palma is undeniably meaningful but I would suggest giving a bit more emphasis to broader implications on the aspects that would make it usable in another region and context.

I think you approach could gain relevance if discussed against similar approaches developed for other natural hazards and for tacking the deriving emergencies, as well as for multi-hazard scenarios. A wider comparative perspective or a discussion linking the results to other active volcanic areas or areas prone to other hazards (e.g. landslides) would enhance the study's impact and utility for a more diverse audience. Expanding the contextualization would help readers better understand how the presented methods and insights could be adapted to different geographic or socioeconomic settings.

**We thank the reviewer for this important observation. In our knowledge there are some studies applying graph indicators to analyse resilience or performance of networks in case of a given disturbance (e.g., tsunami, floods, hurricanes). Averaging indicators have been also used to rank cities or transportation networks in correlation with their population (e.g., Boeing, 2022). However, a detailed analysis and comparison of indicators across time has been rather used in the financial sector to understand systemic relationships in the market institutions before and after economic crises. We have expanded the discussion of these perspectives in lines 813-820.**

**In the absence of any other example in volcanic areas, the same methodology has been tested for the 8 historical eruptions of La Palma by using the actual road network and past lava field footprints (Figure 11), as an example of applicability. A more explicit discussion to other insular contexts (i.e., closed road systems) and continental areas (open systems) has been therefore added in lines 883-885.**

With regards to the methodology, I suggest to discuss a bit more how these ex-post assessments can help future risk management. How can these be included in capacity planning, emergency management and in the definition of future exposure and impact scenarios? I really like your idea of using past lava flow locations and this might be discussed explaining how this could be complemented (e.g. with modeling) to provide future scenarios to land use planners and infrastructure managers. You already started discussing this in line line 907-915 but I think a bit more emphasis could be given to this.

**Many thanks for your insight. This methodology is well structured, rapid to apply, and the python package (OSMnx) is friendly and runs with low computational resources, which could make easy to use for risk management and during crisis management processes. This is particularly important in insular and monogenetic volcanism that involves significant challenges for land-use planning and rapid emergency response due to the reduced space and sudden volcanic changes. We highlighted its potential on future scenarios to inform land use planning and risk reduction (lines 945-950).**

Finally, I think the manuscript could benefit from an overall polishing to avoid repetitions which are sometimes confusing for the reader and perhaps distract from the main message (see specific comments below).

**We polished and deeply reviewed the whole manuscript to avoid redundancies and simplify language when needed.**

Overall, I congratulate the authors for their rigorous and insightful work, which makes a valuable contribution to disaster impact analysis; with the suggested enhancements to broaden its contextual reach, this study has the potential to become a significant reference for the scientific and risk management communities.

**No response needed.**

**Specific comments:**

**Introduction**:

In this work it is somehow unclear why you use the term systemic or cascading impacts, and if you are tackling both. In particular, you mention in line 93-95 that the cascading effects are triggered due to the interaction between roads and other exposed assets, so, to my understanding, we are talking about the induced impacts due to road network disruptions. However, other systemic impacts are also described. I think the reader would benefit for a clarification in the introduction so it's clear what is meant.

**Thank you for your comment. We agree with the reviewer that this terminology is confusing and overlapping across different studies. We specifically use the term "cascading impacts" to describe the nonlinear chains of consequences as a result of complex interactions between systems (as defined by Cutter 2018, also seen as ripple or domino effect by Pescaroli and Alexander, 2015). The complexity of these chains lies on the**

**intricate relationships of multiple dimensions of vulnerability with multiple volcanic hazards. Since it is complicated, we explore a causal order of impacts from physical, to loss of functionality, to systemic impact that are ultimately the result of the associated vulnerabilities. We have modified the text in lines 50-57, 124 and 127-137, hoping that this is clearer now.**

Lines 100-104: It could be worth mentioning the study from Scaini et al. 2014 on the loss of functionality in terms of physical and systemic vulnerability induced by compound volcanic hazards in Tenerife on exposed assets including roads.

**We included this citation in the revised version in Line 56**

Fig.2: The primary road (black line) should be included in the legend, in addition to the caption

**Thanks for this. We suspect you mean to refer to Figure 1B; the legend will be completed in the revised version.**

**Methodology**:

Line 293-294: I think the reader would benefit from a broader description of the involved stakeholders and how they were involved, and for example if the same stakeholders were participating to all the encounters. Also, do the mentioned discussions only include structured interviews or is there also a different kind of interaction (e.g. group meetings)? Did they show you the location of the impacts using a map, or how did they identify the impacted areas and/or the location of the assets? Also, In line with my comment on the generalization of your approach, how do you suggest to adapt the interview questions to different contexts? How was this done passing from Dominguez et al., 2011 to this context?

**We restructured this section substantially to clarify the methodology applied in this research, and particularly the engagement approach undertaken with stakeholders in La Palma. To clarify here, we did not conduct structured interviews, but we followed best-practice guidance for engagement and interactions at the Science-Practice-Policy Interface (SPPI) (Wyborn et al. 2017, Tambe et al. 2023). This was in very large part due to the fact that this study was highly reactive, involving ad hoc meetings with stakeholders that were highly time-limited and operationally busy during the emergency management, response and recovery periods of the 2021 Tajogaite eruption.**

**Designing a participatory research approach requires substantial time and resources, both of which were highly limited for the duration of this study. Future work will follow a prescribed research engagement approach, given we will have the liberty of time and resources post-recovery (following the approach outlined in Weir et al. (2024), Natural Hazards).**

**With the new Methodology structure we clarified the nature of stakeholders interactions, and the strategies adopted in this research in comparison with Dominguez et al., 2021. Specifically, the forensic approach proposed by Dominguez et al. was complemented with a detailed network analysis. The data compiled from stakeholders discussions served to feed the forensic (and cascading patterns) analysis, but also to validate the decisions overtaken for the network analysis (e.g., speed restrictions, Origin-to-Destination points).**

Line 300: I think this sentence explains very well what is done and could be moved to the initial methodological part rather than the data acquisition section.

**We thank the reviewer for this suggestion that has considerably improved the rational of the new section structure.**

Table 1: why 'disrupted' is FL-I and 'reduced' is FL-II? How were these functionality levels defined? Were they all observed in the case-study?

**Levels of functionality were defined in collaboration with stakeholders. We wanted to differentiate between the direct reduced service of road network due to tephra fallout (FL-II) and exclusion zone (FL-III), from the indirect disrupted service due to increasing traffic and disturbances on the network, even when hazards were not present (or insignificant, i.e., in the case of low thickness tephra fallout). The boundaries of these zones were highly discussed, and the majority of stakeholders agreed that traffic disturbances reached as far as Punta Gorda and Santa Cruz (see Figure 6B). This is described in Lines 569-573.**

Line 343-346: The information in the road network data should be moved to the data section

**Thanks for this suggestion. We reorganised this section to add a new sub-section "3.4 GIS setup and data source".**

**Results:**

Section 4.1 mostly presents the impacts in a narrative form and it's hard for the reader to distinguish if these information were collected from the presented methods, or if they are part of the context and the already available information. In the second case, these narratives could be provided at the beginning to explain to the reader what is the situation, while the results should present your original findings.

**This section covers information collected and compiled from discussions with stakeholders, that's why it makes part of the Results section. We believe that after clarifications on the methodology section (discussed above), this will be made clearer.**

Lines 385 -389 and Supplementary material on impacts to other critical infrastructure systems: I am unsure why these impacts are summarized here, if it's to demonstrate the impacts due to the volcanic hazards or to the road closure. This li ks back to the question on the definition of cascading and systemic impacts, which I think could be made clearer in the text.

**It has been already demonstrated that impacts on critical infrastructure involves complex and long cascading chains of effect (Pescaroli and Alexander, 2015; Cutter, 2018); and that we need a system-of-system approach to better quantify it (i.e., systemic vulnerability) (e.g., Rinaldi et al., 2001; Rinaldi et al. 2004; Eusgeld et al. 2011, Weir et al. 2024b). In this particular study we focus on the causal order of impacts starting from the physical damage on the road network, and how this provoked a loss of functionality in the network but also a systemic impact on other sectors. The whole chain is seen as "cascading impact" which is then discretised in several orders of impact. For this reason, we introduced here a brief description on what were the indirect consequences on water supply and agriculture sector. We hope the revised version with correction on the introduction and the methodology section is clearer now.**

Table 5: I think this table is very relevant to understand what are the results of this work.

Row 2, column 5: 'Highly disruptive' is a bit generic

Row 7, column 4: is the surface missing?

**Many thanks for catching this. It has been corrected.**

Section 4.2.2: The initial part is a repetition of the method which should be already explained in the methodology section.

**We adapted the Methods and Results sections to be aligned and avoid repetitions.**

Section 4.2.3: Similarly to section 4.1 there is a broad description of what happened but it is somehow unclear what fraction of this information was gathered using your methods (e.g. interviews) and which information was already available in advance. Even in this second case, if you reinterpreted this based on the impact chains that you identified and on your methods, this can be part of the results, but it should be made clear to the reader.

**The results presented here are the outcomes of discussions with stakeholders based on the engagement approach we have followed. This has been substantially clarified in the Methods section.**

**Discussion:**

Line 730-740: I would include in the introduction some background on the ex-post analyses and the approach in the field of disaster risk reduction, with literature references (e.g. AM Ferreira et al., 2023; F. Atun, 2024), in order to highlight the relevance of you work beyond the case-study. Also, this could be discussed to show what are the improvements of the approach that you propose.

**Thanks for this observation and references. We have emphasized the utility of PEIAs, and forensic approaches, in particular, for disaster risk reduction as it informs future pre-event risk analysis and planning. Due to space and word constraints, we cannot expand beyond 1-2 sentences, but we included Ferreira et al. 2023 as a key reference on this topic in the discussion (line 701).**

Line 745-746: 'identify the most impactful cascading effects': Is this shown in the figures? Or was this part of the replies that they gave you to the interviews?

**Stakeholders help identifying that the major cascading patterns were associated with the impact to the road network. We have however modified to "major cascading" since this study focus *only* in one pattern (line 711). However, social or economic impacts were also very important. Besides, Fig. 2, Fig. 6 and Table 5, show a summary of all these aspects.**

Line 760-764: It would be relevant to explain what kind of data would support a deeper and more extended deductive/inductive reasoning. This could help other researchers interested in applying this tool in other areas where such data might be available.

**In these lines, we discuss the data that was available (specified in lines 510-515, see below), the gaps we found indeed, and how we explore the occurrence of different functionality scenarios. The deductive/inductive reasoning lies from defining specific questions with the data we account for (event tree in Fig. 7). In a case study where more data is available, the probabilities associated with each scenario could eventually be calculated. In the absence of this data, what we propose here is to quantify these functionality scenarios through the quantification of network indicators across time.**

Line 765-768: This sentence is not super clear to me, could you rephrase? Does it mean that, since the majority of the tephra deposition happened in the first phase and that the impacted road length are almost constant during the following phases?

**It has been rephrased it as follows (lines 728-731),**

*"Concerning the reduced service due to tephra accumulation (FL-II), we found that almost the totality of the surface estimated from the whole tephra deposit at the end of the eruption (>190 km$^2$) was affected since the beginning of the eruption, resulting in an almost constant road length for FL-II impact, regardless of the amount of accumulated tephra (Fig. 7B). This means that the few millimeters covering the surface immediately implied a FL-II, however, a detailed evolution of tephra thickness over time is required to assess different levels of impact."*

**(This then introduces the next paragraph, where FL-II are then sub-divided into categories in relation with tephra thickness).**

Given that you comment on different CI (e.g. water networks) and given the presence of the water reservoirs that are, both here and in other Canary Islands, so relevant for local activities, I would suggest adding some references in the introduction about the exposure and vulnerability of these (e.g. Stewart et al., 2006, 2009).

**We thank the reviewer for this suggestion. The aim of analysing water disruption was focalised as an indirect impact starting from road network damage and disruption. We could eventually analyse another cascading pattern starting from the physical impacts on water reservoirs and networks that has its own complexities; and as you said, very relevant and complicated in all Canary Islands. However, this is beyond the present study.**

Fig. 6C: Is it systemic or cascading?

**Figure 6C corresponds to the systemic impact in a cascading chain where a causal order of physical-functional-systemic has been established. We believe with the modifications in the introduction and methods sections this is clearer now.**

Line 814-815: how much of this analysis can be done a priori? e.g. analyzing the existing network and its potential pitfalls using a combination of scenario modeling and network analysis?

**Network analysis can be performed any time to analyse the performance of a system. Your suggestion of combining this with scenario modelling has been emphasized in Lines 824-827.**

Apart from the road type (and associated speed), did you account for the road path (e.g. if it is mostly straight or curved) and the pavement type? Also, how different could this scenario in case of different weather conditions (e.g. under heavy rainfalls), especially in areas served by unpaved secondary or tertiary roads?

**With the available information, our study accounts for 3 different weights (length, travel speed and travel time) and we explore the different outcomes for these variables. Analysis was then under the umbrella of road type (i.e., primary, secondary, tertiary) to define different speeds, hence, indirectly accounting for pavement type. However, future**

**analysis including traffic, slope or pavement type could add more real insights on the impedance of roads. This is mentioned in Lines 804-807.**

Do you foresee to tackle the lack of traffic data in some other way (e.g. using phone data, such as in the work of Yabe et al., 2022; Wilson et al., 2016; Giardini et al., 2023)? Or, if not, shall this be addressed in the future?

**This is a very interesting strategy that could be used for future eruptions. One of the major limitations we have during crisis is the lack of time and existing guidelines to collect impact data, and we are not likely to be able to acquire this data so far after the event. The proposed strategy would have a great utility. However, accounting for traffic fluxes requires adopting a different strategy to model accessibility (e.g., agent-based models; see Bonadonna et al., 2022). Such a modelling strategy requires a more complex initialisation that would prevent the ease of reproducibility compared to the network-based method adopted here.**

**Conclusions:**

Following my comment on the results, I think the conclusions are a bit generic and could be made stronger by giving more emphasis to the specific results achieved with the analysis (e.g. from table 5, fig. 10-11). In particular, from Fig. 10 it seems that the travel time increases a lot for R1, and this might set a priority for emergency managers.

**We emphasized the utility of this research specifically in La Palma. However, the result of Fig. 10 for the Route 1 – emergency is very specific and highly dependent to the Tajogaite scenario. Indeed, the monogenetic nature of La Palma could totally modify the expected impact scenarios of future events.**

**Minor comments:**

If possible, I would suggest to translate the tables in S2 into English

**Absolutely agree. The reviewed Supplementary material includes both, the original alert levels of PEVOLCA, and a complementary English language version.**

I'm not a big fan of acronyms and sometimes I feel their use could be reduced in favor of readability.

**We agree that generally the use of acronyms should be restricted. Here we really tried to use the least acronyms possible to convey the necessary details, and to avoid confusion we added a list of abbreviations at the beginning of the manuscript.**

**Supplement:**

In the section regarding air traffic disruptions, I think the Sara et al. Reference Barsotti et al.:

Sara, B., Simona, S., Giovanni, M., Alicia, F., Aline, P., Georgios, V., de Zeeuw van Dalfsen, E., Lars, O., Adriano, P., Jean-Christophe, K., Susan, L., Rita, C., Mauro, C., Jordane, C., Charlotte, V. B., Mauro, D. V., de Chabalier, J. B., Teresa, F., Fontaine Fabrice, R., Arnaud, L., Rui, M., Joana, M., Roberto, M., Anne, P. M., Jean-Marie, S., Ivan, V., Kristín, V., Samantha, E., and Giuseppe, S.: The European Volcano Observatories and their use of the aviation colour code system, Springer Berlin Heidelberg, https://doi.org/10.1007/s00445-024-01712-0, 2024

**Thanks for catching this, indeed the reference was there but not in the bibliography. Corrected.**

Also, on the tephra-induced impacts there are more references that could be provided.

**Relevant references were added through addressing previous comments.**

With regards to the selection and classification of exposure assets (S3), I think it should be discussed and compared with other existing classification and taxonomies (e.g. the GED4ALL) and their adoption for exposure assessment in other study areas. Exposure is inherently multi-hazard, so the approach would benefit from the adoption of a more general taxonomy. Also, I suggest discussing how local-scale methods such as the one proposed here can be integrated in the overall multi-hazard risk assessment.

**We thank the reviewer for this remark and agree that a more general taxonomy scheme would benefit of any multi-hazard research. However, changing the exposure classification scheme is not feasible at this point in the present study unfortunately, but would be considered for future research. However, a discussion of exposure assets, classification and taxonomies, or the integration of local-scale methods in the multi-hazard risk assessment go beyond the scope of this study.**
* * *
**Reviewer #2**

**Title: Quantifying cascading impacts through road network analysis in an insular volcanic setting: the 2021 Tajogaite eruption of La Palma Island (Spain)**

This study by Dominguez et al. presents a comprehensive post-event impact assessment of the 2021 Tajogaite eruption on La Palma Island, Spain. The authors apply an interesting approach of combining a forensic impact assessment analysis with network analysis to evaluate and quantify functionality loss and systemic impacts to infrastructure from the eruption. I think this will make an important and useful contribution within the scope of NHESS and to the academic literature on post-event impact assessment. However, there are a few issues that require consideration that I outline below.

*No response needed.*

**Ambiguity between observed and modelled impacts and effects:** It can be unclear in places whether an impact of effect described in the paper was observed/reported by stakeholders or derived from the modelling approach. This is because the text shifts at times between describing what happened (or was reported to have happened) and then what is inferred through the modelling. The differences can get a bit lost in dense paragraphs. I think the manuscript would benefit greatly from greater clarification. Perhaps through the use of a table. This would aid the reader to interpret the added value of using the network analysis approach.

**We thank the reviewer for this insight. We substantially modified the Methods section to widely clarify the impact data source, the stakeholders engagement approach and the expected results from network analysis. For clarification here, the PEIA proposed coupled two strategies: first, a forensic approach based on a causal order to describe cascading impacts: physical – functional – systemic impacts (following Dominguez et al., 2021). Second, a graph network analysis to quantify the functional impact (using graph**

indicators) and  systemic impact (through travel time between different sectors). We modified the text to clarify how the observed/reported impacts from stakeholders have been used mainly for the forensic analysis, but also to validate the results from graph indicators due to the real 2021 lava and tephra occurrence. We would like then to emphasize that rather than a *modelling* approach, we present a *network analysis* to study past impacts.

**Clarifying and critically evaluating the graph-theory approach among existing geospatial methods:** The manuscript would benefit from a clearer articulation of what value graph-theory indicators add as the indicators are somewhat theoretical. The authors state this is pioneering in volcanology, but I think that is possibly too strong of a claim. Whilst this appears to be the first time researchers have applied topological indicators in this way for a volcano case study, it is not the first time that network analysis or geospatial analysis has been used to supplement observed impact information for post-event impact assessment for volcanic eruptions. As the authors highlight, EBC and ECC are both toplogical measures and so do not necessarily reflect functional aspects of road importance (e.g., traffic capacity, importance to critical facilities etc). It would be valuable for the authors to critically evaluate the implications of this, and to situate this approach in relation to other geospatial methods that have been applied in post-event impact assessments for volcanic eruptions. The authors do have a section on potential and limitations, but I think this currently falls short of the necessary critique of the method, its added value, and how it might fit with other previously used approaches.

In the introduction of the manuscript, we highlighted pioneering studies in volcanology that applied graph networks to the quantification of vulnerability, such as Weir et al. (2024) and Mossoux et al. (2019). Our intention was not to present our study as the pioneering work, but rather to emphasize that, to the best of our knowledge, this is the first time topological network indicators have been applied specifically within the field of volcanology.

We have included in the references what we believe represents the state-of-the-art regarding the application of graph theory and network modelling in the context of volcanic impacts and risk. Should the reviewer have additional suggestions, we would gladly include them.

We also appreciate the valuable suggestion to expand Section 5.2 on the potential and limitations of our approach. In response, we included a more comprehensive discussion of the implications of using graph-based indicators, particularly in comparison with other methodologies used for volcanic post-event impact or risk assessments.

**Ethics:** The authors highlight that they undertook discussions with stakeholders following the eruption. Can they please include a statement somewhere indicating that the necessary ethics approvals (if required) for their institution for this type of research were obtained. Often this can go in the acknowledgements or methodology section.

Thank you for your comment. we followed best-practice guidance for engagement and interactions at the Science-Practice-Policy Interface (SPPI) (Cash et al. 2003; Wyborn et al. 2017, Tambe et al. 2023). Interactions were semi-structured, involving several recurring discussion points ensuring a mutual beneficial and highly adaptive process. No formal interviews were conducted, no individuals have been named, nor have their personal opinions been shared. The nature of the engagement did not reach the minimum

conditions needed for requiring ethics approval at the University of Geneva, where the funding was held and the majority of the authors were based.

However, an ethics approval is in place for the large project that funded this study. This funding project (SNSF #188757) is principally concerned with volcanic risk in the Central and Southern Andean Volcanic Zones and hence is tailored to engagement undertaken in South America with local research partners. However, the ethics approval granted for the funding project outlines a robust ethical approach for engagement with volcanic risk management stakeholders that heavily informed this study in La Palma. *We* provided mention in the acknowledgements and note that it is the ethics approval for the project that funded this study.

**Some smaller points**

- The manuscript in places uses a lot of jargon that could be more plain spoken (or at least clear definitions provided early in the manuscript). Examples include: insular environment/context, centrality indicators, "compound lava", single annular primary network, system performance phases named as disturbance progress, degraded, and restorative states.

We thank the reviewer for this comment. We carefully verified the whole manuscript with special attention to remove any unnecessary jargon and provide definitions of unclear terms. Specifically,

- Insular environment refers strictly to the geographical disconnection of the mainland, i.e., islands (https://archive.espon.eu/islands-and-insularity)

- Definitions of centrality indicators are provided in Table 3.

- Compound hazards definition is provided in Line 53

- For precision, annular road network has been replaced by ring topology with associated reference (Line 399)

- Concerning CI performance phases, there is quite a lot of debate in terminology about performance and resilience of networks. As stated in the manuscript (lines 780-785), here we adopted the phases of disturbance progress, degraded state and restorative state proposed for electricity and gas supply disruptions.

- There are a number of overly verbose and structurally complex sentences throughout. As one example: "Further research should focus on the selection of indicators (and parameterisation) with a physical meaning, which must be developed as we explore better this approach, by applying to future on-going eruptions (syn- and post-event impact assessments) or to future potential scenarios (pre-event impact assessments)." This sentence would benefit from being split into a couple sentences. I suggest the authors review the entire manuscript to reduce redudant words and complex sentences.

Thanks for this comment. We revised the language and complicate sentences by addressing comments of both reviewers.

The authors mention that they assume no clean-up occurs as there is no information they can use for this. This is reasonable, but as the authors point out this will have a significant impact on the outputs and inferences made. Could the authors explore end-member scenarios similar to what was done for speed restrictions?

**This is a very important aspect. In fact, in the preliminary stages of this research, we wanted to explore the effect of clean-up operations, as was done, for example, in the case of impact on buildings where a record of clean-up operations was available (see Reyes-Hardy et al. 2024, https://www.frontiersin.org/journals/earth-science/articles/10.3389/feart.2023.1303330/full).**

**It is important to note that clean-up operations are crucial in the case of roof collapse of buildings. However, in the case of roads, our results clearly showed that the major component influencing the evolution of graph indicators is the lava field. This means that the effect of tephra fallout only narrowly dispersed the value of indicators, but the crucial trend was defined by the lava field evolution (see, for example, Fig. 9B and 9C). This is clearly stated in lines 648–650.**

**For this reason, we considered that introducing fictitious scenarios of clean-up operations of roads would introduce an extra source of uncertainty that is totally difficult to validate (since we do not have any record of when and where the clean-up operations were done), and in the end, the indicator value would approximate (if not equal to) the lava scenario (brown triangles in Figure 9B, 9C), as it would be the case of tephra = 0.**

- Line 112: "under the light of network analysis" is strange phrasing and unclear.

**We will change for "in light of network analysis"**

- Line 278: Why was the decision unprecedented?

**With "unprecedented" we mean that the decision of allowing residents to enter into the exclusion zone to collect their belongs was never done before in any emergency that Spain has faced. This was stated by emergency stakeholders, particularly civil protection practitioners. We consider that it is pertinent to mention this decision in our manuscript not only because of its relevance in the crisis management but also for its implications on the road network accessibility and connectivity. Indeed, this measure provoked long traffic jams and important disruptions of the road network not directly impacted by lava or tephra.**

- A few more sentences to expand upon the rationale for the selection of origin-destination locations. It's stated that these are key selected locations, but why?

**We thank the reviewer to point out this. In fact, we wanted to demonstrate with few examples the effects on other systems given the loss of connectivity of the road network. To do that, we chose 4 sectors: emergency, health, agriculture, education and we carefully selected the coordinates of locations that were relevant for the communities and stakeholders to show the changes on travel time before, during and after the eruption of Tajogaite. We clarified this in Lines 362-365.**

- Line 402-404: Is the sentence relating to local and global newspapers relevant? Not clear to me.

**Media and the community in general were quite concerned about the lava reaching the sea. This event is relevant to our focus on the road network because on that day (29 September), the western flank of the island was disconnected (evidenced by all graph indicators dropping, Figure 10). Media and emergency stakeholders emphasized this event and date. Besides, there was particular concern about the secondary explosions that lava could produce upon contact with water. Indeed, some confinement measures were implemented due to poor air quality (Fig. 2).**

Line 414: How much ash fell at the port? Can you specify a thickness?

**Unfortunately, there is no official measurement of thickness at the port. Practitioners only mention that "some ash" fell down few days during the eruption. Direct impacts due to tephra were no reported.**

- Line 430: reference to Fig. 2B. Should this be 3B? Fig 2 doesn't have a B.

**Totally correct. Thanks for pointing this.**

- Table 5: What does "highly disruptive" mean in this context under the Tephra column?

**We wanted to emphasize that although tephra has not direct physical damage on roads, it could be highly disruptive. We will modify by "disruption due to reduced visibility, loss of traction, covering of marks, reduction of skid resistance and total covering, depending on deposit thickness"**

- Line 509: What level of damage occurred? Was this cosmetic or functional damage?

**It was physical damage including cracks, deep scratches and abrasion of markings. We added these specific physical impacts on the manuscript (line 485)**

- Line 623-624: why are percentage ranges provided? Isn't clear what the percentages refer to.

**It is provided in percentage as a total measure of timing affectation during the scholar year. We considered more easily to understand and compare with other regions/countries, given the highly variable scholar number of days.**

- Line 780: "IS1b till the end" shoud this be "until the end"?

**Modified**

I would like to finish by applauding the authors on producing what I believe is an interesting piece of research and a valuable and timely contribution. It is important to document the impacts of volcanic eruptions and the authors have taken this a step further by providing an additional tool and approach to explore post-event impact assessment for volcanic eruptions.

**We kindly thank the reviewer for her/his valuable insights in this review. We believe all the comments addressed here will contribute to improve our manuscript.**

---

## Referee Report (RR1)

Dear authors,
thank you very much for all the work done on the article.

My only suggestion is regarding the sentence in lines 1067-1069, I still think it's a bit unclear, and I suggest rephrasing a bit, similarly to this (if I understood correctly what you wanted to say): 'Concerning the reduced service due to tephra accumulation (FL-II), we found that even a few millimeters of tephra were enough to affect nearly the entire area eventually covered by the full deposit (>190 km²) from the very beginning of the eruption. As a result, the road length impacted by FL-II remained almost constant throughout the event, regardless of the total accumulated tephra (Fig. 7B). This indicates that a minimal tephra load was sufficient to trigger FL-II conditions. However, a detailed analysis of tephra thickness evolution over time is needed to assess varying levels of impact.'

Thank you very much for your detailed replies to my comments and questions. I appreciate the effort you put into addressing the feedback and congratulate you for the excellent work done.

---

## Author Response (AR2)

Dear authors,
thank you very much for all the work done on the article.

My only suggestion is regarding the sentence in lines 1067-1069, I still think it's a bit unclear, and I suggest rephrasing a bit, similarly to this (if I understood correctly what you wanted to say): 'Concerning the reduced service due to tephra accumulation (FL-II), we found that even a few millimeters of tephra were enough to affect nearly the entire area eventually covered by the full deposit (>190 km²) from the very beginning of the eruption. As a result, the road length impacted by FL-II remained almost constant throughout the event, regardless of the total accumulated tephra (Fig. 7B). This indicates that a minimal tephra load was sufficient to trigger FL-II conditions. However, a detailed analysis of tephra thickness evolution over time is needed to assess varying levels of impact.'

Thank you very much for your detailed replies to my comments and questions. I appreciate the effort you put into addressing the feedback and congratulate you for the excellent work done.

**The lines have been modified accordingly to this suggestion (now lines 728-734). Additionally I slightly modified Fig 1A to better show the location of Spain peninsula and Canary archipelago.**
**Lastly, I added the name of Jonathan Lemus to acknowledgements.**